# UProp: Investigating the Uncertainty Propagation of LLMs in Multi-Step Decision-Making

## Abstract

As Large Language Models (LLMs) are integrated into safety-critical applications involving sequential decision-making in the real world, it is essential to know when to trust LLM decisions. Existing LLM Uncertainty Quantification (UQ) methods are primarily designed for single-turn question-answering formats, resulting in multi-step decision-making scenarios, e.g., LLM agentic system, being underexplored. In this paper, we introduce a principled, information-theoretic framework that decomposes LLM sequential decision uncertainty into two parts: (i) internal uncertainty intrinsic to the current decision, which is focused on existing UQ methods, and (ii) extrinsic uncertainty, a Mutual-Information (MI) quantity describing how much uncertainty should be inherited from preceding decisions. We then propose `UProp`, an efficient and effective extrinsic uncertainty estimator that converts the direct estimation of MI to the estimation of Pointwise Mutual Information (PMI) over multiple Trajectory-Dependent Decision Processes (TDPs). `UProp` is evaluated over extensive multi-step decision-making benchmarks, e.g., AgentBench and HotpotQA, with state-of-the-art LLMs, e.g., GPT-4.1 and DeepSeek-V3. Experimental results demonstrate that `UProp` significantly outperforms existing single-turn UQ baselines equipped with thoughtful aggregation strategies. Moreover, we provide a comprehensive analysis of `UProp`, including sampling efficiency, potential applications, and intermediate uncertainty propagation, to demonstrate its effectiveness.

## 1 Introduction

Large Language Models (LLMs) (Zhao et al., 2023) are increasingly deployed in real-world applications that involve sequential decision-making, such as Agentic AI (Wang et al., 2024b), where LLMs interact with environments across multiple steps. Many of these applications, including multi-round medical consultations (Zhou et al., 2023) and autonomous robotic control (Zeng et al., 2023; Duan et al., 2022), are safety-critical. Given that LLMs are prone to hallucinations and errors (Huang et al., 2025), it is crucial to assess the reliability of their decisions and understand when these decisions can be trusted. Uncertainty quantification (UQ) estimates the degree of uncertainty or lack of confidence that a model has in its predictions, essentially reflecting how unsure it is about the "correctness" of its output (Gawlikowski et al., 2023). It has proven to be a promising method for quantifying the reliability of LLM decisions, such as in hallucination detection and correction (Yin et al., 2024).

Current LLM UQ methods primarily focus on single-step question-answering tasks (Malinin & Gales, 2020), where LLMs are expected to respond to a query. These methods (Kuhn et al., 2023; Duan et al., 2024a; Lin et al., 2024b; Qiu & Miikkulainen, 2024) quantify uncertainty by measuring the semantic diversity of LLM output space. While these "single-step" methods offer reliable uncertainty estimations at each step, in the multi-step decision-making scenarios, they fail to capture the propagation of uncertainty within a decision trajectory. SAUP (Zhao et al., 2024) trains a Hidden Markov Model (HMM) to predict the aggregation weights of per-step uncertainty within a decision trajectory. However, it requires the ground-truth labels from the test domain and does not investigate uncertainty propagation in a principled manner. In this paper, we study ***how the uncertainty of the current decision should be influenced by preceding decisions?***

Figure 1: The pipeline of uncertainty propagation of LLMs in multi-step agentic decision-making.

We approach LLM multi-step decision-making from a Bayesian perspective and develop an information-theoretic framework to analyze its uncertainty propagation. Specifically, we decompose the LLM's uncertainty at each decision step into (i) **Intrinsic Uncertainty** (IU), which reflects the internal uncertainty dependent solely on the current state, e.g., observation the LLM is facing, and (ii) **Extrinsic Uncertainty** (EU), which represents the uncertainty introduced by (or "inherited from") the variability of preceding decisions. Among the two components, intrinsic uncertainty can be reliably estimated, as it is convenient to sample from the decision distribution from LLMs for uncertainty estimation (Malinin & Gales, 2020). In contrast, estimating extrinsic uncertainty is more challenging because it is described as the Mutual Information (MI) between the current decision distribution and each of the preceding decision distributions (Cover, 1999). This becomes intractable as MI necessitates the decision distributions at each step of the reasoning process. Even from a Monte Carlo (MC) sampling perspective, the multi-step decision making demands the exploration of an exponentially expanding decision space (Kraskov et al., 2004), which is computationally infeasible.

We propose UProp, an efficient and effective estimator of extrinsic uncertainty. In general, UProp complies with the MC approximation idea, which first samples decision processes from the decision space and then estimates per-process uncertainty propagation. Specifically, UProp (1) first conducts Trajectory-Dependent Decision Process (TDP) sampling from the exponential decision space: each TDP sample results in a complete decision trajectory (from beginning to end) along with multiple samples at each step. (2) Then, for each TDP, UProp estimates the uncertainty propagation by approximating the more feasible Pointwise MI (PMI). With convergence analysis, we prove that the Trajectory-Dependent PMI approximation converges to the actual MI in the LLM multi-step decision-making scenario, under a mild local smoothness assumption.

UProp is evaluated in extensive LLM multi-step decision-making scenarios including the Operating System Agent split in AgentBench (Liu et al., 2023) and multi-hop benchmarks such as HotpotQA (Yang et al., 2018) and StrategyQA (Geva et al., 2021) bound with a Wikipedia engine, over powerful LLMs, such as GPT-4.1-Nano (Achiam et al., 2023), GPT-3.5-Turbo (Brown et al., 2020), and DeepSeek-V3 (DeepSeek-AI et al., 2024). We compare UProp with state-of-the-art single-turn UQ methods, including Semantic Entropy (SE) (Kuhn et al., 2023), Deg (Lin et al., 2024b), SAR (Duan et al., 2024a), Semantic Density (SD) (Qiu & Miikkulainen, 2024), G-NLL (Aichberger et al., 2025), etc., equipped with thoughtful step aggregation strategies. Experimental results demonstrate that UProp significantly outperforms these baselines (by 2.3% ∼ 11% AUROC). We further characterize UProp from the perspective of sampling efficiency, selective prediction, and intermediate uncertainty propagation. Our results indicate that extrinsic uncertainty plays an important role in the uncertainty quantification of LLM sequential decision-making. Our contributions are:

- We provide an information-theoretic framework that decomposes the uncertainty of LLM sequential decision into intrinsic and extrinsic uncertainty. We highlight the necessity of propagating extrinsic uncertainty along the LLM decision chain for more accurate uncertainty quantification.

- We provide UProp, an efficient and effective extrinsic uncertainty estimator. UProp approximates the Mutual Information (MI) between decision distributions by expecting the Pointwise Mutual Information (PMI) among trajectory-dependent samplings.

- UProp is evaluated over extensive LLM sequential decision-making scenarios, involving powerful LLMs and state-of-the-art baseline methods. Experimental results demonstrate that UProp significantly outperforms best-performing baselines in LLM multi-step decision-making scenarios.

## 2 PRELIMINARY

### 2.1 UNCERTAINTY QUANTIFICATION IN AUTO-REGRESSIVE GENERATIONS

From the Bayesian perspective, UQ measures the uncertainty within the predictive probability distribution $p_{\boldsymbol{\theta}}(\boldsymbol{y}|\boldsymbol{x})$ over the LLM output space $\mathcal{Y}$, given a parameterized LLM $f_{\boldsymbol{\theta}}$ and instruction $\boldsymbol{x}$. One of the most popular UQ methods is quantifying the total uncertainty of the predictive distribution (Gawlikowski et al., 2023) by calculating its Predictive Entropy (PE). However, considering that the analytic form of LLM predictive distributions is intractable, i.e., do not have access to all possible $|V|^k$ $k$-length generations in the LLM output space (where $V$ is the vocabulary size), a more convenient way is approximating via Monte-Carlo (MC) sampling (Malinin & Gales, 2020):

$$PE(\boldsymbol{x}) = H(\boldsymbol{y}|\boldsymbol{x}) = \int p_{\boldsymbol{\theta}}(\boldsymbol{y}|\boldsymbol{x}) \log(p_{\boldsymbol{\theta}}(\boldsymbol{y}|\boldsymbol{x})) \, d\boldsymbol{y} \approx -\frac{1}{N} \sum_i^N \log p_{\boldsymbol{\theta}}(\boldsymbol{y}^{(i)}|\boldsymbol{x}), \ \boldsymbol{y}^{(i)} \sim p_{\boldsymbol{\theta}}(\boldsymbol{y}|\boldsymbol{x}),$$

where $N$ is the number of samples and $p_{\boldsymbol{\theta}}(\boldsymbol{y}^{(i)}|\boldsymbol{x}) = \prod_i^{L_i} p_{\boldsymbol{\theta}}(z_i|z_{<i}, \boldsymbol{x})$ is the generative probability of $\boldsymbol{y}^{(i)}$ with length $L_i$. $z_i$ is the $i$-th token of $\boldsymbol{y}^{(i)}$. Length-normalization is also commonly applied to mitigate the length sensitivity: $LN\text{-}PE(\boldsymbol{x}) \approx -\frac{1}{N} \sum_i^N \frac{1}{L_i} \log p_{\boldsymbol{\theta}}(\boldsymbol{y}^{(i)}|\boldsymbol{x})$. Furthermore, Kuhn et al. (2023) proposes that PE may overestimate output uncertainty due to the existence of semantic clusters, i.e., different generations may share the same semantics. To mitigate this, Semantic Entropy (SE) calculates the cluster-wise predictive entropy with MC approximation:

$$SE(\boldsymbol{x}) \approx -\frac{1}{C} \sum_i^C \log(p_{\boldsymbol{\theta}}(\boldsymbol{c}_i|\boldsymbol{x})), \ p_{\boldsymbol{\theta}}(\boldsymbol{c}_i|\boldsymbol{x}) = \sum_{\boldsymbol{y} \in \boldsymbol{c}_i} p_{\boldsymbol{\theta}}(\boldsymbol{y}|\boldsymbol{x}),$$

where $C$ is the number of semantic clusters and $\boldsymbol{c}_i$ is the $i$-th cluster consisting of generations $\boldsymbol{y}_i$ sharing the same semantics. Following the semantic consistency, a series of UQ methods, including Deg (Lin et al., 2024b), SAR (Duan et al., 2024a), and SD (Qiu & Miikkulainen, 2024), are proposed.

### 2.2 LLM MULTI-STEP AGENTIC DECISION-MAKING

LLM multi-step agentic decision-making (Liu et al., 2023; Duan et al., 2024b) is usually modeled as a stochastic Markov Decision Process (MDP) $(f_{\boldsymbol{\theta}}, \mathcal{O}, \mathcal{Y}, \mathcal{T})$, where LLM $f_{\boldsymbol{\theta}}$ interacts with the environment continuously. $\mathcal{O}$ and $\mathcal{Y}$ are observation space and decision space, respectively. $\mathcal{T} : \mathcal{Y}^* \to \mathcal{O}$ is the deterministic observation transition function of the environment, where $\mathcal{Y}^*$ denotes a finite sequence of decisions. Assume at the $t$-th decision step, the decision $\boldsymbol{y}_t \in \mathcal{Y}$ is sampled by

$$\boldsymbol{y}_t \sim p_{\boldsymbol{\theta}}(\boldsymbol{y}_t|\boldsymbol{o}_{t-1}, \boldsymbol{y}_{t-1}, \cdots, \boldsymbol{o}_1, \boldsymbol{y}_1, \boldsymbol{x}),$$

where $\boldsymbol{o}_i \in \mathcal{O}$ is the observation at $i$-th step and $\boldsymbol{x}$ is the instruction. We assume the observation transition function is deterministic when the preceding decisions $\mathcal{Y}^* = [\boldsymbol{y}_i]^{t-1}$ are determined, i.e., the decision distribution $\boldsymbol{y}_t$ is solely dependent on preceding decisions. Thus, we omit all the observation conditions in the following notations, i.e., $\boldsymbol{y}_t \sim p_{\boldsymbol{\theta}}(\boldsymbol{y}_t|\boldsymbol{y}_{1:t-1}, \boldsymbol{x})$.

## 3 METHODOLOGY

### 3.1 PREDICTIVE UNCERTAINTY PROPAGATION IN LLM MULTI-STEP DECISION-MAKING

In the LLM multi-step decision-making process, UQ quantifies the uncertainty within the predictive distribution $p_{\boldsymbol{\theta}}(\boldsymbol{y}|\boldsymbol{x})$. Without loss of generality, we quantify the uncertainty at the $t$-th step predictive distribution $\boldsymbol{y}_t \sim p_{\boldsymbol{\theta}}(\boldsymbol{y}_t|\boldsymbol{x})$. By marginalizing preceding decisions, we obtain the following decomposition (see Section A.1 for detailed procedures):

$$
\begin{aligned}
p_{\boldsymbol{\theta}}(\boldsymbol{y}_t|\boldsymbol{x}) &= \int p_{\boldsymbol{\theta}}(\boldsymbol{y}_t|\boldsymbol{y}_{t-1}, \boldsymbol{x}) p(\boldsymbol{y}_{t-1}|\boldsymbol{x}) d\boldsymbol{y}_{t-1} \\
&= \int \underbrace{p_{\boldsymbol{\theta}}(\boldsymbol{y}_t|\boldsymbol{y}_{1:t-1}, \boldsymbol{x})}_{\text{Intrinsic Uncertainty}} \underbrace{\prod_i^{t-1} p_{\boldsymbol{\theta}}(\boldsymbol{y}_i|\boldsymbol{y}_{1:i-1}, \boldsymbol{x})}_{\text{Extrinsic Uncertainty}} d\boldsymbol{y}_1 d\boldsymbol{y}_2 \cdots d\boldsymbol{y}_{i-1}.
\end{aligned}
\tag{1}
$$

We show that the total uncertainty at step $t$ could be described in **Intrinsic Uncertainty** (IU) and **Extrinsic Uncertainty** (EU): (1) IU refers to the expected variance of $\boldsymbol{y}_t$ given all preceding decisions, i.e., $\mathbb{E}_{\boldsymbol{y}_{1:t-1}}\big[\mathrm{Var}_{\boldsymbol{y}_t}(\boldsymbol{y}_t|\boldsymbol{y}_{1:t-1}, \boldsymbol{x})\big]$. It captures the uncertainty within the predictive distribution itself and corresponds to what "single-step" UQ methods typically estimate; (2) EU quantifies the variance of $\boldsymbol{y}_t$ introduced by prior decisions, expressed as $\mathrm{Var}_{\boldsymbol{y}_{1:t-1}}\big(\mathbb{E}_{\boldsymbol{y}_t}[\boldsymbol{y}_t|\boldsymbol{y}_{1:t-1}, \boldsymbol{x}]\big)$, which is the uncertainty that should be propagated from preceding decisions.

By the chain rule of conditional entropy, entropy $H(\boldsymbol{y}_t|\boldsymbol{x})$ could be expressed as the following (see Section A.2 for detailed procedures):

$$H(\boldsymbol{y}_t|\boldsymbol{x}) = \mathbb{E}_{\boldsymbol{y}_{1:t-1}\sim p(\boldsymbol{y}_{1:t-1}|\boldsymbol{x})}[H(\boldsymbol{y}_t|\boldsymbol{y}_{1:t-1}, \boldsymbol{x})] + \sum_i^{t-1}\left(H(\boldsymbol{y}_t|\boldsymbol{x}) - H(\boldsymbol{y}_t|\boldsymbol{y}_i, \boldsymbol{x})\right)$$

$$= \underbrace{\mathbb{E}_{\boldsymbol{y}_{1:t-1}\sim p(\boldsymbol{y}_{1:t-1}|\boldsymbol{x})}[H(\boldsymbol{y}_t|\boldsymbol{y}_{1:t-1}, \boldsymbol{x})]}_{\text{Intrinsic Uncertainty}} + \underbrace{\sum_i^{t-1} I(\boldsymbol{y}_t; \boldsymbol{y}_i|\boldsymbol{y}_{i+1:t-1}, \boldsymbol{x})}_{\text{Extrinsic Uncertainty}}, \tag{2}$$

where $I(\boldsymbol{y}_t; \boldsymbol{y}_i|\boldsymbol{y}_{i+1:t-1}, \boldsymbol{x})$ is Mutual Information (MI). The total uncertainty of the decision process $\mathcal{P} = (\boldsymbol{y}_1, \boldsymbol{y}_2, \cdots, \boldsymbol{y}_t) \sim p_{\boldsymbol{\theta}}(\mathcal{P}|\boldsymbol{x})$ becomes:

$$H(\mathcal{P}) = \mathbb{E}_{P\sim\mathcal{P}}\Big[-\log p_{\boldsymbol{\theta}}(P|\boldsymbol{x})\Big] = \mathbb{E}_{P\sim\mathcal{P}}\Big[-\sum_i \log p_{\boldsymbol{\theta}}(\boldsymbol{y}_i|\boldsymbol{y}_{1:i-1}, \boldsymbol{x})\Big], \tag{3}$$

Within the decomposition in Equation (2),

- Intrinsic Uncertainty could be conveniently MC approximated by first sampling multiple generations from $p_{\boldsymbol{\theta}}(\boldsymbol{y}_t|\boldsymbol{y}_{1:t-1}, \boldsymbol{x})$ and then aggregating with existing algorithms, such as PE, SE, and SAR.

- Extrinsic Uncertainty is characterized by the cumulative MI between $\boldsymbol{y}_t$ and all preceding decisions $\boldsymbol{y}_{1:t-1}$. It reflects the extent to which uncertainty in $\boldsymbol{y}_t$ is reduced as each prior decision is resolved, i.e., *knowledge uncertainty* (Malinin, 2019). In this sense, extrinsic uncertainty quantifies the degree of "increased determinism" in $\boldsymbol{y}_t$ that arises from conditioning on $\boldsymbol{y}_{1:t-1}$.

However, directly calculating extrinsic uncertainty is intractable, as it requires an awareness of predictive distributions. Even from the perspective of MC approximation or density estimation, the estimation of $I(\boldsymbol{y}_t; \boldsymbol{y}_i|\boldsymbol{y}_{i+1:t-1}, \boldsymbol{x})$ is still challenging as it necessarily explores an exponentially spanned decision space: outer sampling from preceding decision distributions $\boldsymbol{y}_{<t}$ with inner sampling from $\boldsymbol{y}_t$. Moreover, in the LLM decision-making scenarios, this exponential interaction with the environment becomes harder to afford. Please refer to Section A.3 for more discussion.

### 3.2 UPROP: ESTIMATE EXTRINSIC UNCERTAINTY WITH TRAJECTORY-DEPENDENT POINTWISE MI

We propose UProp as an efficient and effective estimator of EU. In general, UProp complies with the MC approximation idea, which first samples decision processes from the decision space and then estimates per-process uncertainty propagation. Specifically, it converts the direct estimation of MI to the estimation of *Pointwise Mutual Information* over *Trajectory-Dependent Decision Processes*:

**Trajectory-Dependent Decision Process (TDP) Sampling**  Starting from the beginning decision step ($t = 1$), we first sample $N$ decisions $\{\boldsymbol{y}_t^{(1)}, \boldsymbol{y}_t^{(2)}, \cdots, \boldsymbol{y}_t^{(N)}\} \sim p_{\boldsymbol{\theta}}(\boldsymbol{y}_t|\boldsymbol{x})$; then, we randomly select one sample $\boldsymbol{y}_t^{(k)}$ by probability, as the preceding realization of the $(t + 1)$-th step; then, we sample $N$ decisions from $\boldsymbol{y}_{t+1} \sim p_{\boldsymbol{\theta}}(\boldsymbol{y}_{t+1}|\boldsymbol{y}_{1:t} = \boldsymbol{y}_{1:t}^{(k)}, \boldsymbol{x})$. We repeat this protocol until $\boldsymbol{y}_T^{(k)}$ achieves an end decision at step $T$, e.g., the decision to return the final answer. In this way, each TDP will be expressed as:

$$TDP_z = \{<\boldsymbol{y}_1^{(k)}, \{\boldsymbol{y}_1^{(n)}\}_{n,n\neq k}^N >, <\boldsymbol{y}_2^{(k)}, \{\boldsymbol{y}_2^{(n)}\}_{n,n\neq k}^N >, \cdots, <\boldsymbol{y}_T^{(k)}, \{\boldsymbol{y}_T^{(n)}\}_{n,n\neq k}^N >\},$$

consisting of one complete decision trajectory: $\{\boldsymbol{y}_1^{(k)}, \boldsymbol{y}_2^{(k)}, \cdots, \boldsymbol{y}_T^{(k)}\}$, and multiple samplings conditioned on preceding trajectories at each step: $\{\{\boldsymbol{y}_1^{(n)}\}_{n,n\neq k}^N, \{\boldsymbol{y}_2^{(n)}\}_{n,n\neq k}^N, \cdots, \{\boldsymbol{y}_T^{(n)}\}_{n,n\neq k}^N\}$.

**Pointwise Mutual Information (PMI) in TDP**  We study the uncertainty propagation over TDPs. Conditioned on the realizations within TDP, MI $I(\boldsymbol{y}_t; \boldsymbol{y}_{t-1}|\boldsymbol{x})$ over TDP at step $t$ becomes a PMI:

$$PMI(\boldsymbol{y}_t; \boldsymbol{y}_{t-1} = \boldsymbol{y}_{t-1}^{(k)}|\boldsymbol{x}) = D_{KL}(p_{\boldsymbol{\theta}}(\boldsymbol{y}_t|\boldsymbol{y}_{t-1}^{(k)}, \boldsymbol{x}) \parallel p_{\boldsymbol{\theta}}(\boldsymbol{y}_t|\boldsymbol{x})). \tag{4}$$

Then, combining Equations (2) and (4) the MC approximated total uncertainty of TDP, $\mathcal{P}_{TDP} \sim p_{\boldsymbol{\theta}}(\mathcal{P}_{TDP}|\boldsymbol{x})$, becomes

$$H(\mathcal{P}_{TDP}) \approx \frac{1}{Z} \sum_z^Z \sum_t^{T_z} \Big( H(\boldsymbol{y}_t|\boldsymbol{y}_{1:t-1}^{(k)}, \boldsymbol{x}) + \sum_i^{t-1} PMI(\boldsymbol{y}_t; \boldsymbol{y}_i^{(k)}|\boldsymbol{y}_{i+1:t-1}^{(k)}, \boldsymbol{x}) \Big), \qquad (5)$$

where $Z$ is the sampling number of TDP and $T_z$ is the number of steps within the $z$-th TDP, i.e., the length of the $z$-th TDP's decision trajectory.

**Theorem 1** *(Convergence of the TDP Sampling) With sufficiently large TDP sampling, the total uncertainty of TDP converges to the total uncertainty $H(\mathcal{P})$ (Equation (3)): $H(\mathcal{P}_{TDP}) \to H(\mathcal{P})$, when $Z \to \infty$.*

Please refer to Section A.4 for the proof of Theorem 1. Instead of directly estimating MI over the exponential decision space, UProp first samples linear-spanning decision processes and then uses the more feasible PMI over each TDP as the approximation of the total uncertainty $H(\mathcal{P})$.

### 3.3 SPREADING DECISION DISTRIBUTIONS BY PRECEDING VARIANCE

Given a TDP $P_z$, we consider approximating PMI by spreading from the known conditional distribution, under a mild local smoothness assumption. Specifically, without loss of generality, we consider the MC approximation of $PMI(\boldsymbol{y}_t; \boldsymbol{y}_{t-1}^{(k)}|\boldsymbol{x})$:

$$PMI(\boldsymbol{y}_t; \boldsymbol{y}_{t-1} = \boldsymbol{y}_{t-1}^{(k)}|\boldsymbol{x}) = \mathbb{E}_{\boldsymbol{y}_t} \Big[ \log \frac{p_{\boldsymbol{\theta}}(\boldsymbol{y}_t|\boldsymbol{y}_{t-1}^{(k)}, \boldsymbol{x})}{p_{\boldsymbol{\theta}}(\boldsymbol{y}_t|\boldsymbol{x})} \Big] \approx \frac{1}{N} \sum_n^N \log \frac{p_{\boldsymbol{\theta}}(\boldsymbol{y}_t^{(n)}|\boldsymbol{y}_{t-1}^{(k)}, \boldsymbol{x})}{p_{\boldsymbol{\theta}}(\boldsymbol{y}_t^{(n)}|\boldsymbol{x})}, \quad (6)$$

where $\boldsymbol{y}_t^{(n)}$ is the $n$-th sample from TDP's $t$-step samples and $p_{\boldsymbol{\theta}}(\boldsymbol{y}_t^{(n)}|\boldsymbol{y}_{t-1}^{(k)}, \boldsymbol{x})$ is calculated and saved during TDP sampling. In terms of $p_{\boldsymbol{\theta}}(\boldsymbol{y}_t^{(n)}|\boldsymbol{x})$, we approximate it by spreading the preceding semantic variance with "neighborhood-weighted" average:

$$\hat{p}_{\boldsymbol{\theta}}(\boldsymbol{y}_t|\boldsymbol{x}) = \sum_n^N p_{\boldsymbol{\theta}}(\boldsymbol{y}_t|\boldsymbol{y}_{t-1}^{(k)}, \boldsymbol{x}) \cdot K_N(d(\boldsymbol{y}_{t-1}^{(n)}, \boldsymbol{y}_{t-1}^{(k)})), \qquad (7)$$

where $K_\tau(x) = \big( \frac{1}{\sqrt{2\pi}} e^{(-\frac{x^2}{2})} \big)^\tau$ is a Gaussian Kernel with $\tau$ controls its sharpness. $d(\boldsymbol{y}_1, \boldsymbol{y}_2)$ is a distance measurement between the two decisions. We take $s = N$ to highlight those samples close to $\boldsymbol{y}_{t-1}^{(k)}$, i.e., the extrinsic uncertainty is dominated by its surroundings. In this way, the PMI is approximated as:

$$\widehat{PMI}(\boldsymbol{y}_t; \boldsymbol{y}_{t-1}^{(k)}|\boldsymbol{x}) = \frac{1}{N} \sum_n^N \log \frac{p_{\boldsymbol{\theta}}(\boldsymbol{y}_t^{(n)}|\boldsymbol{y}_{t-1}^{(k)}, \boldsymbol{x})}{\hat{p}_{\boldsymbol{\theta}}(\boldsymbol{y}_t^{(n)}|\boldsymbol{x})} = -\log \sum_n^N K_N(d(\boldsymbol{y}_{t-1}^{(n)}, \boldsymbol{y}_{t-1}^{(k)})). \qquad (8)$$

Heuristically, spreading by preceding semantic variance indicates that a low-uncertainty preceding decision distribution introduces less uncertainty to the current step. Extremely, a degenerate preceding distribution introduces no uncertainty to follow-up decisions. It is worth noting that Equation (7) simplifies the propagation of uncertainty by using a neighborhood-weighted average, which primarily captures local similarities in the prior step decision space. In Section A.6, we illustrate that this design choice is both principled and practical.

**Theorem 2** *(Convergence of the PMI Approximation) Assume that $p_{\boldsymbol{\theta}}(\boldsymbol{y}_t|\boldsymbol{y}_{t-1}, \boldsymbol{x})$ satisfies a local smoothness with respect to $\boldsymbol{y}_{t-1}$, i.e., for any fixed context $\boldsymbol{x}$, there exists a sufficiently small neighborhood around $\boldsymbol{y}_{t-1}$ such that for all points $\boldsymbol{y}_{t-1}'$ within this neighborhood:*

$$\forall \epsilon > 0, \exists \beta > 0 : |\boldsymbol{y}_{t-1} - \boldsymbol{y}_{t-1}'| < \beta, \ \text{then } |p_{\boldsymbol{\theta}}(\boldsymbol{y}_t|\boldsymbol{y}_{t-1}, \boldsymbol{x}) - p_{\boldsymbol{\theta}}(\boldsymbol{y}_t|\boldsymbol{y}_{t-1}', \boldsymbol{x})| < \epsilon.$$

*Then, the PMI estimation (Equation (8)) spreading from the preceding variance converges to the actual MC approximation of PMI (Equation (4)): $\widehat{PMI}(\boldsymbol{y}_t|\boldsymbol{y}_{t-1}, \boldsymbol{x}) \to PMI(\boldsymbol{y}_t|\boldsymbol{y}_{t-1}, \boldsymbol{x}).$*

Please refer to Section A.5 for the proof of Theorem 2 and further discussion. The local smoothness assumption is natural and practical and has been widely conducted in existing LLM analysis (Malinovskii et al., 2024). Combining the total uncertainty convergence (Theorem 1) and PMI convergence (Theorem 2), the total uncertainty (Equation (3)) approximation is derived (combining Equations (5) and (8)) as:

$$H(\mathcal{P}) = H(\mathcal{P}|\boldsymbol{x}) \approx H(\mathcal{P}_{TDP}|\boldsymbol{x}) = \frac{1}{Z}\sum_{z}^{Z}\frac{1}{\lambda_z}\sum_{t}^{T_z}\left(H(\boldsymbol{y}_t|\boldsymbol{y}_{1:t-1}^{(k)},\boldsymbol{x}) + \sum_{i}^{t-1}\widehat{PMI}(\boldsymbol{y}_t;\boldsymbol{y}_i^{(k)}|\boldsymbol{y}_{i+1:t-1}^{(k)},\boldsymbol{x})\right),$$
(9)

where $\frac{1}{\lambda_z}$ is an additional "step length-normalization" item:

**Step Length-Normalization** Similar to "length-normalization" (Malinin & Gales, 2020), due to the accumulation over $\widehat{PMI}(\boldsymbol{y}_t;\boldsymbol{y}_i^{(k)}|\boldsymbol{y}_{i+1:t-1}^{(k)},\boldsymbol{x})$, Equation (9) implies the Step Length Bias: *longer decision steps encodes higher extrinsic uncertainty*. The total uncertainty of a TDP is normalized by $\lambda_z = \sum_t^{T_z}\sigma_t = \sum_t^{T_z}(1 + \frac{EU}{IU}) = T_z + \sum_t^{T_z}\frac{\sum_i^{t-1}\widehat{PMI}(\boldsymbol{y}_t;\boldsymbol{y}_i^{(k)}|\boldsymbol{y}_{i+1:t-1}^{(k)},\boldsymbol{x})}{H(\boldsymbol{y}_t|\boldsymbol{y}_{1:t-1}^{(k)},\boldsymbol{x})}$, where $\sigma_t$ indicates the relative inflation of the uncertainty at step $t$ due to extrinsic contributions. In this way, the step bias is mitigated, and different TDPs with varying lengths become comparable.

Equation (9) estimates the overall uncertainty of decision distributions $\mathcal{P}$. However, in some scenarios, e.g., hallucination detection, one may care more about the uncertainty of a specific prediction $\boldsymbol{y}^*$, i.e., the uncertainty of the maximum probability class. Given model output $\boldsymbol{y}^*$, e.g., the greedy generation, its uncertainty could be approximated as:

$$H(\boldsymbol{y}^*|\boldsymbol{x}) = H(\mathcal{P}_{\boldsymbol{y}^*}|\boldsymbol{x}) \approx H(\mathcal{P}_{TDP,\boldsymbol{y}^*}|\boldsymbol{x}),$$
(10)

where $\mathcal{P}_{\boldsymbol{y}^*}$ is a decision process distribution consisting of decision processes ending with decision $\boldsymbol{y}^*$, and $\mathcal{P}_{TDP,\boldsymbol{y}^*}$ is a TDP distribution consisting of TDPs ending with decision $\boldsymbol{y}^*$.

In our implementation, we calculate PE as the estimation of intrinsic uncertainty. In the rest of this paper, we denote `UProp` to be $H(\mathcal{P}_{TDP,\boldsymbol{y}^*}|\boldsymbol{x})$ by default. For decision distance measurement $d$, we use the simple string fuzzy matching from thefuzz (SeatGeek, 2020) as the distance measurement. In Section A.7, we provide further discussion and comparison to other alternatives such as Natural Language Inference (NLI) (He et al., 2020).

## 4 EXPERIMENTS

### 4.1 EXPERIMENTAL SETUP

**Environments and Benchmarks** We evaluate `UProp` over both multi-step decision-making and multi-step reasoning benchmarks:

- **Multi-Step Decision-Making**: we consider AgentBench-OS, the *Operating System (OS)* Agent benchmark in *AgentBench* (Liu et al., 2023). In AgentBench-OS, the LLM Agent is instructed to finish a task by interacting with a Linux OS, e.g., *find an executable file named echo-love* (see Section B.1 for demonstrations and prompt templates).

- **Multi-Step Reasoning**: we consider the popular multi-hop question-answering benchmarks: *HotpotQA* (Yang et al., 2018) and *StrategyQA* (Geva et al., 2021). In these benchmarks, LLMs are tasked to answer a question requiring multi-hop reasoning. LLM is prompted in a ReAct (Yao et al., 2022) style: *Reasoning-Action-Observation*, where each action will provide a keyword to a Wikipedia engine for retrieval (see Section B.2 for demonstrations and prompt templates).

**LLMs and Sampling** We consider state-of-the-art commercial LLMs (GPT-4.1-Nano-2025-04-14 (Achiam et al., 2023), GPT-3.5-Turbo-0125 (Brown et al., 2020)) and open-source LLMs (QWen2.5-72b-Instruct (Yang et al., 2024), DeepSeek-V3 (DeepSeek-AI et al., 2024), and Gemma-2-27b-it (Riviere et al., 2024)) as backbones. For generative hyperparameters, we use greedy search to generate responses for correctness evaluation and multinomial search with a temperature set to 0.8 for MC sampling. For all the generations, we set the maximum number of new tokens to be 512. By default, the trajectory sample number $Z$ and the per-step sample number $N$ are set to 10.

**UQ Baselines** We consider 7 popular single-step LLM UQ methods: Perplexity (PPL), Lexical Similarity (LS) (Fomicheva et al., 2020), PE (Malinin & Gales, 2020), SE (Kuhn et al., 2023),

Table 1: AUROC results over AgentBench-Operating System and StrategyQA benchmarks. For single-turn baseline UQ methods, uncertainties are aggregated by ***averaging*** over all steps.

| Models | Success Rate | PPL | LS | PE | SE | Deg | G-NLL | SD | sentSAR | UProp (ours) |
|---|---|---|---|---|---|---|---|---|---|---|
| **Benchmark**: AgentBench-Operating System | | | | | | | | | | |
| GPT-4.1-Nano | 0.307 | 0.725 | 0.756 | 0.768 | 0.770 | 0.757 | 0.763 | _0.779_ | 0.775 | **0.781** |
| GPT-3.5-Turbo | 0.275 | 0.747 | 0.750 | _0.782_ | 0.765 | 0.765 | 0.745 | 0.749 | 0.777 | **0.791** |
| Gemma-2-27b-it | 0.289 | 0.747 | 0.636 | 0.760 | 0.755 | 0.652 | _0.787_ | 0.766 | 0.755 | **0.814** |
| DeepSeek-V3 | 0.310 | 0.729 | 0.636 | 0.724 | 0.716 | 0.655 | **0.767** | 0.717 | 0.722 | **0.767** |
| Qwen2.5-72B-Instruct | 0.508 | 0.625 | 0.620 | **0.707** | 0.687 | 0.631 | 0.671 | 0.678 | 0.678 | _0.704_ |
| **Average** | 0.338 | 0.715 | 0.679 | _0.748_ | 0.738 | 0.692 | 0.747 | 0.738 | 0.741 | **0.771** |
| **Benchmark**: StrategyQA | | | | | | | | | | |
| GPT-4.1-Nano | 0.691 | 0.512 | 0.492 | _0.542_ | 0.503 | 0.528 | 0.502 | 0.499 | 0.527 | **0.544** |
| GPT-3.5-Turbo | 0.611 | 0.593 | 0.438 | **0.623** | _0.611_ | 0.440 | 0.608 | 0.600 | 0.607 | 0.604 |
| Gemma-2-27b-it | 0.777 | 0.698 | 0.615 | 0.669 | 0.624 | 0.622 | _0.759_ | 0.640 | 0.667 | **0.766** |
| DeepSeek-V3 | 0.790 | 0.573 | 0.548 | 0.559 | 0.558 | 0.575 | _0.583_ | 0.574 | 0.563 | **0.607** |
| Qwen2.5-72B-Instruct | 0.796 | 0.500 | 0.495 | _0.573_ | _0.573_ | 0.493 | 0.556 | 0.567 | 0.563 | **0.617** |
| **Average** | 0.733 | 0.575 | 0.518 | _0.593_ | 0.574 | 0.526 | 0.606 | 0.576 | 0.585 | **0.628** |

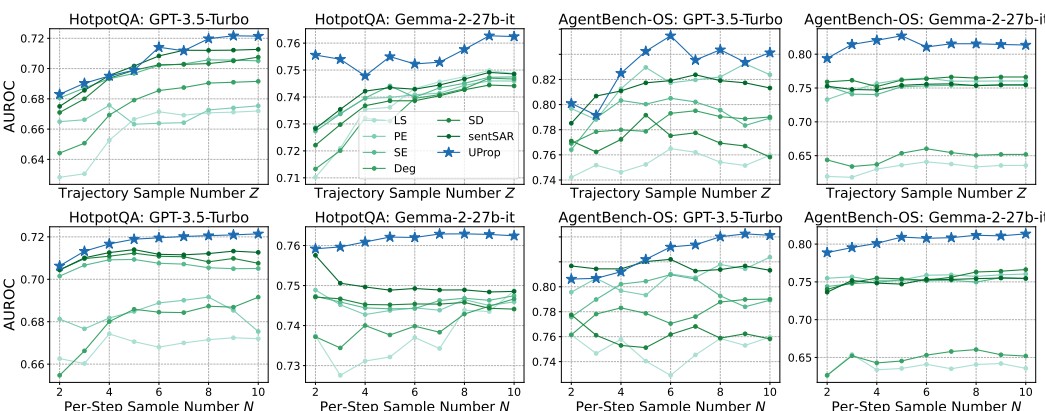

Figure 2: Comparing the sampling efficiency of UProp with baselines.

Deg (Lin et al., 2024b), SD (Qiu & Miikkulainen, 2024), sentSAR (Duan et al., 2024a), and G-NLL (Aichberger et al., 2025). For a fair comparison and also a straightforward adaptation, baselines are calculated over the same TDP samples. Specifically, for each TDP sample, baseline methods first (1) calculate per-step uncertainty by their design. Then, (2) the TDP total uncertainty is aggregating all these per-step uncertainties by either *average* or *Root Mean Square (RMS)*. Eventually, (3) the final uncertainty is the averaging of TDPs' total uncertainties. Apart from these UQ baselines, in Section D, we present **broader adaptions** that compare UProp with baselines using a single greedy trajectory or last decision only to quantify uncertainty, with alternative naive baselines such as mean (or max) token entropy.

**Evaluation Metric** Following existing work (Kuhn et al., 2023) in this domain, we evaluate UQ by assessing how well it predicts the correctness of the model's generated answers for a given question, with the metric Area Under the Receiver Operating Characteristic Curve (AUROC). In Section C.1, we also provide the hallucination detection performance of UProp and baselines evaluated by accuracy and F1.

## 4.2 PERFORMANCE ON MULTI-STEP DECISION-MAKING BENCHMARKS

We report the general performance (Success Rate) of LLMs and the AUROC of baselines and UProp, over AgentBench-Operating System (OS) and StrategyQA datasets. The performance of baselines aggregated by *averaging* is reported in Table 1 (please refer to Section C.2 for RMS aggregation comparison). It is shown that UProp achieves the best UQ performance in most settings, compared to both average and RMS aggregation. It significantly outperforms existing methods in general, e.g., UProp outperforms baselines by 2.3% ~ 9.2% AUROC in AgentBench-OS and 3.5% ~ 11% AUROC in StrategyQA.

## 4.3 SAMPLING EFFICIENCY

To quantify the sampling efficiency, we compare the AUROC of sampling-based baselines and UProp over various TDP sampling numbers, i.e., $Z \in [2, 10]$, and per-step sampling numbers within each TDP, i.e., $N \in [2, 10]$ (we only vary one of the sampling numbers at each time and fix the other sampling numbers to be 10). Results are summarized in Figure 2. It is shown that UProp outperforms baselines in most sampling configurations, including when very few trajectory samplings or per-step samplings are available. It implies that UProp is effective and efficient in LLM multi-step UQ.

## 4.4 ABLATION STUDY

**IU vs. EU** We investigate the effectiveness of IU and EU individually. In Table 2, we provide the AUROC when removing each of these components from UProp, over the AgentBench-OS benchmark. In general, both IU and EU contribute to the performance improvement.

Table 2: Ablation study of IU and EU in UProp.

| Model | UProp | w/o EU | w/o IU |
|---|---|---|---|
| GPT-4.1-Nano | **0.781** | 0.726 (**-5.5%**) | 0.770 (-1.1%) |
| GPT-3.5-Turbo | **0.791** | 0.747 (-4.4%) | 0.717 (**-7.4%**) |
| Gemma-2-27b-it | **0.813** | 0.765 (**-4.8%**) | 0.794 (-1.9%) |
| DeepSeek-V3 | **0.767** | 0.700 (**-6.7%**) | 0.733 (-3.4%) |
| Qwen2.5-72B-Instruct | **0.704** | 0.652 (**-5.2%**) | 0.684 (-2.0%) |

However, w/o EU brings larger performance drops than IU, indicating that EU is an essential component in uncertainty quantification.

**Selective Prediction** Rejecting response by uncertainty is an important UQ application, e.g., hallucination detection in LLMs. In Table 3, we report the selective prediction performance comparison, evaluated by metric Area Under Accuracy-Rejection Curve (AUARC) (Nadeem et al., 2009) over the StrategyQA benchmark. We show that UProp substantially outperforms baselines in most cases, e.g., UProp outperforms baselines by up to 2.6% AUARC. This indicates that UProp retains better performance in rejecting incorrect answers.

Table 3: The evaluation of selective prediction with AUARC.

| Models | PPL | LS | PE | SE | Deg | sentSAR | SD | UProp |
|---|---|---|---|---|---|---|---|---|
| GPT-4.1-Nano | 67.2 | 62.9 | 68.2 | 66.6 | 63.8 | 67.2 | 66.7 | **68.5** |
| GPT-3.5-Turbo | 64.1 | 54.0 | 67.2 | 65.1 | 54.1 | 64.6 | 64.6 | 66.8 |
| Gemma-2-27b-it | 84.5 | 79.7 | 83.2 | 81.8 | 80.2 | 83.1 | 82.2 | **86.0** |
| DeepSeek-V3 | 77.5 | 76.1 | 78.6 | 78.4 | 77.9 | 78.6 | 79.2 | **79.7** |
| Qwen2.5-72B-Instruct | 74.1 | 75.7 | 78.4 | 78.5 | 76.2 | 78.2 | 77.5 | **81.1** |
| **Average** | 73.5 | 69.7 | 75.1 | 74.1 | 70.4 | 74.3 | 74.0 | **76.4** |

**Uncertainty as Correctness Indicator** Uncertainty could serve as the indicator of correctness, which is one of the potential applications of UQ. We study the effectiveness of baselines and UProp in identifying correct answers from multiple generations. Specifically, for each question, we first sample 10 generations and then select the one with the lowest uncertainty (estimated by various UQ methods) as the final answer. We calculate the general performance, i.e., success rate (SR), of these final answers. We conduct this experiment over AgentBench-OS and the results are reported in Figure 3. We show that UQ effectively improves SR, and UProp achieves the best performance among all the baselines.

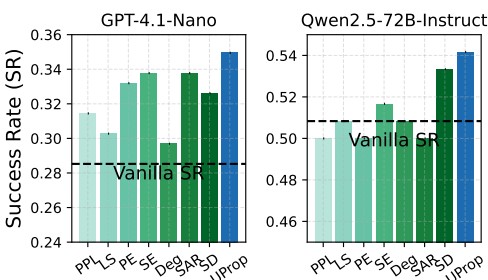

Figure 3: Uncertainty as the correctness indicator for improved LLM performance.

**EU Significantly Correlates to Correctness** To quantify the utility of the introduced EU, we calculate the estimated EU for each decision-making trajectory and provide the correctness vs. EU scatters in Figure 4. A stronger negative correlation indicates that EU is an effective estimation of the correctness of trajectories. We conduct experiments with Gemma-2-27b-it, DeepSeek-V3, and QWen2.5-72B-Instruct over the AgentBench-OS task. It is shown that EU is significantly correlated to correctness, demonstrating the practical utility of EU and UProp.

**UProp in Long-Step Decision-Making** To compare the performance of UProp when dealing with short and long trajectories, we report the Excess AUARC Geifman et al. at each trajectory group. The reason we choose Excess AUARC rather than AUROC is that (1) longer trajectories are inherently more challenging than shorter ones. This variation in difficulty places AUROC values on incomparable scales across trajectory groups; (2) Excess AUARC calculates the pure gain by

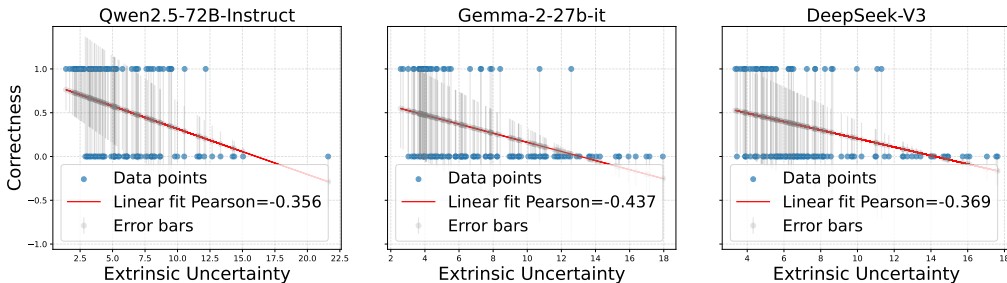

Figure 4: Quantify the correlation between EU and correctness. It is shown that the EU is negatively correlated with the correctness, indicating that the introduced uncertainty propagation effectively estimates the correctness of LLM trajectories.

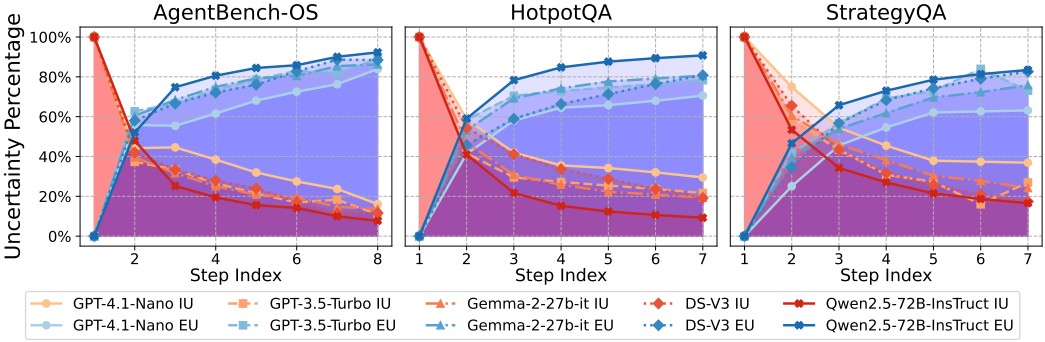

Figure 5: The percentage of intrinsic and extrinsic uncertainty at each step. The Red shadow area is the percentage of IU and the Blue shadow area is the percentage of EU.

rejecting uncertain answers, which is a fair evaluation metric regarding the utility of UQ methods. Please refer to Section C.4 for more details. Results are summarized in Table 4. It is shown that Excess AUARC is stable at different numbers of steps, indicating that UProp is still effective on questions requiring longer trajectories.

### 4.5 INTERMEDIATE UNCERTAINTY PROPAGATION ANALYSIS

To understand how uncertainty is propagated along decision trajectories and identify the contributions of IU and EU individually, we provide the uncertainty percentage of IU and EU, i.e., $\frac{IU}{IU+EU}$ and $\frac{EU}{IU+EU}$, at each decision step. Results are summarized in Figure 5 (the detailed per-model results are provided in Section C.3). We observe that (1) IU usually contributes significantly to the first few decision steps while EU

Table 4: Evaluate `UProp` in long-step trajectories with Excess AUARC.

| Models | 2 Steps | 6 Steps | 10 Steps |
|---|---|---|---|
| Gemma-2-27b-it | 0.098 | 0.043 | 0.062 |
| DeepSeek-V3 | 0.037 | 0.031 | 0.029 |

dominates the rest. As the decision step grows, EU heavily affects the total uncertainty of the decision step, highlighting the significance of EU in uncertainty propagation; (2) GPT-4.1-Nano embraces a relatively smaller EU percentage compared to other LLMs at the later decision steps, e.g., EU and IU share closer percentages at the end step in StrategyQA. This indicates that GPT-4.1-Nano has more stable and less uncertain decisions.

## 5 RELATED WORK

**Uncertainty Quantification (UQ) of LLMs** **Uncertainty Quantification (UQ) of LLMs** In LLMs, UQ quantifies the uncertainty within its prediction distribution (Malinin & Gales, 2020; Aichberger et al., 2025). From the perspective of entropy, uncertainty could be measured by the log probability of generations sampled from the output space (Gawlikowski et al., 2023). However,

entropy may overestimate uncertainty due to the semantic clusters. To address this issue, Semantic Entropy (SE) (Kuhn et al., 2023) clusters LLM outputs by semantics and then calculates cluster-wise entropy as the uncertainty. Deg (Lin et al., 2024b) is specifically designed for black-box UQ and it models output consistency by either node connectivity or eigenvalues of a semantic graph (which is further extended by INSIDE (Chen et al., 2024) in LLM hidden space). SAR (Duan et al., 2024a) reveals token-level and sentence-level semantic imbalance in LLM UQ. The token-level semantic importance is further extended by CSL (Lin et al., 2024a). Semantic Density (SD) (Qiu & Miikkulainen, 2024) calculates the density of a target generation within a semantic space as the uncertainty. KLE (Nikitin et al., 2024) encodes semantic similarities of LLM outputs to mitigate the "semantic overlapping" among semantic clusters.

**LLM Multi-Step Decision-making** It refers to sequential interactions between an LLM agent and its environment (Wang et al., 2024b), spanning OS (Liu et al., 2023), Wikipedia, games (Duan et al., 2024b), and robotics (Liu et al., 2024). Frameworks like ReAct (Yao et al., 2023) introduce a think-act-observe loop, extended by Reflection (Shinn et al., 2023) and Rest (Aksitov et al., 2023) with self-reflection (Ji et al., 2023). $Q^*$ (Wang et al., 2024a) incorporates deliberative planning, while auxiliary modules (graphs (Wu et al., 2025), tools (Paranjape et al., 2023)) enhance reasoning. Stepwise reasoning (Wang et al., 2025) further improves performance by referencing underused information and reducing redundancy. While PlanU (Deng et al.) addresses uncertainty, it functions primarily as a planning algorithm, utilizing MCTS and quantile regression to maximize task rewards and guide exploration. In contrast, our approach is grounded in information theory, aiming not to guide search but to provide a scalar reliability metric specifically for selective prediction and the rejection of incorrect answers.

# 6 CONCLUSION

In this paper, we investigate the uncertainty propagation of LLMs in multi-step decision-making. Specifically, we first provide a principled, information-theoretical framework that decomposes the uncertainty into intrinsic uncertainty and extrinsic uncertainty. We then propose `UProp`, as an efficient and effective estimator of extrinsic uncertainty. We conduct experiments over popular sequential decision-making scenarios and experimental results demonstrate the superior performance of `UProp` compared to best-performing baselines. We further study the intermediate states of `UProp`, such as the performance of `UProp` on long-step trajectories, the percentage of IU and EU at each step, as well as the Pearson correlation between EU and the correctness of trajectories.

**Limitations & Social Impacts** The proposed `UProp` relies on MC sampling for MI estimation. On the one hand, the estimation might be deviated due to insufficient sampling and unknown distribution from the LLM decision space. Moreover, sampling may result in latency in real-world deployment. Also, our study involves closed-source commercial LLMs such as GPT-4.1 and GPT-3.5-Turbo, which may suffer from reproducibility issues due to the continuous updating of these models. We investigate the uncertainty quantification in LLMs, which is one of the most important topics in trustworthy LLMs and responsible LLMs. We expect our method to improve hallucination detection in LLM sequential decision-making and could be used to correct LLM behaviors in uncertain decision scenarios.

**Ethics Statement** While our study does not involve human subjects or sensitive personal data, we acknowledge ethical considerations regarding the deployment of LLMs in real-world environments such as medical consultation or robotics. Our method, UProp, is designed to improve uncertainty estimation in LLMs. When applied in safety-critical domains, its use should be accompanied by human oversight to avoid potential misuse. It can be used to quantify the confidence of LLMs regarding its response, detect hallucination, as well as correct LLM behaviors.

**Reproducibility Statement** In Section 4.1 we provide detailed descriptions of datasets and benchmarks (AgentBench-OS, HotpotQA, and StrategyQA), model backbones (GPT-4.1, GPT-3.5-Turbo, DeepSeek-V3, Qwen2.5, Gemma-2-27B-IT), and evaluation metrics (AUROC, AUARC). The hallucination detection accuracy and F1 results are further included in Section C.1. Complete prompt templates and data processing pipelines are provided in Section B.1–B.2. All codes and configuration scripts will be released upon the final decision of the paper to facilitate reproducibility

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

# A  IMPLEMENTATION DETAILS AND RELATED PROOFS

In this section, we provide detailed proofs and procedures used in this paper. We will reuse the notations we defined before.

## A.1  UNCERTAINTY DECOMPOSITION

The joint distribution for a sequence of events $\boldsymbol{y}_{1:t}$, conditioned on $x$, follows the chain rule for conditional probability:

$$p_{\boldsymbol{\theta}}(\boldsymbol{y}_{1:t}|\boldsymbol{x}) = p_{\boldsymbol{\theta}}(\boldsymbol{y}_1|\boldsymbol{x})p_{\boldsymbol{\theta}}(\boldsymbol{y}_2|\boldsymbol{y},1\boldsymbol{x})p_{\boldsymbol{\theta}}(\boldsymbol{y}_3|\boldsymbol{y}_{1:2}\boldsymbol{x})\cdots p_{\boldsymbol{\theta}}(\boldsymbol{y}_t|\boldsymbol{y}_{1:t-1}\boldsymbol{x}). \quad (11)$$

By marginalizing out previous decisions $\boldsymbol{y}_{1:t-1}$, the distribution $p_{\boldsymbol{\theta}}(\boldsymbol{y}_t|\boldsymbol{x})$ becomes:

$$
\begin{aligned}
p_{\boldsymbol{\theta}}(\boldsymbol{y}_t|\boldsymbol{x}) &= \int p_{\boldsymbol{\theta}}(\boldsymbol{y}_t|\boldsymbol{y}_{1:t-1},\boldsymbol{x})p_{\boldsymbol{\theta}}(\boldsymbol{y}_{1:t-1}|\boldsymbol{x})d\boldsymbol{y}_{1:t-1} \\
&= \int p_{\boldsymbol{\theta}}(\boldsymbol{y}_t|\boldsymbol{y}_{1:t-1},\boldsymbol{x})\prod_i^{t-1}p_{\boldsymbol{\theta}}(\boldsymbol{y}_i|\boldsymbol{y}_{1:i-1},\boldsymbol{x})d\boldsymbol{y}_1 d\boldsymbol{y}_2\cdots d\boldsymbol{y}_{i-1}
\end{aligned}
\quad (12)
$$

## A.2  ENTROPY DECOMPOSITION

Based on conditional mutual information and iteratively applying it to each preceding decision $\boldsymbol{y}_i$, we have the following decomposition:

$$
\begin{aligned}
H(\boldsymbol{y}_t|\boldsymbol{x}) &= H(\boldsymbol{y}_t|\boldsymbol{y}_{t-1},\boldsymbol{x}) + I(\boldsymbol{y}_t;\boldsymbol{y}_{t-1}|\boldsymbol{x}) \\
&= H(\boldsymbol{y}_t|\boldsymbol{y}_{t-2:t-1},\boldsymbol{x}) + I(\boldsymbol{y}_t;\boldsymbol{y}_{t-2}|\boldsymbol{y}_{t-1},\boldsymbol{x}) + I(\boldsymbol{y}_t;\boldsymbol{y}_{t-1}|\boldsymbol{x}) \\
&\cdots \\
&= H(\boldsymbol{y}_t|\boldsymbol{y}_{1:t-1},\boldsymbol{x}) + \sum_i^{t-1}I(\boldsymbol{y}_t;\boldsymbol{y}_i|\boldsymbol{y}_{i+1:t-1},\boldsymbol{x}).
\end{aligned}
\quad (13)
$$

## A.3  MI CALCULATION NECESSITIES EXPONENTIAL EXPLORATION

In a multi-step decision-making process, we denote by $\mathcal{A}$ the decision space at each step, the MI between the $n$-th step distribution $\boldsymbol{y}_n$ and the $m$-th step distribution $\boldsymbol{y}_m$ $(m > n)$, i.e., $I(\boldsymbol{y}_m;\boldsymbol{y}_n)$ requires the joint distribution $p_{\boldsymbol{\theta}}(\boldsymbol{y}_m,\boldsymbol{y}_n|\boldsymbol{x})$:

$$p_{\boldsymbol{\theta}}(\boldsymbol{y}_m,\boldsymbol{y}_n|\boldsymbol{x}) = \int\cdots\int_{\mathcal{A}^{m-n-1}} p_{\boldsymbol{\theta}}(\boldsymbol{y}_1,\boldsymbol{y}_2,\cdots,\boldsymbol{y}_t,\cdots,\boldsymbol{y}_m|\boldsymbol{x})\prod_{k\in\{1,\cdots,m\}\setminus\{n.m\}} d\boldsymbol{y}_k. \quad (14)$$

Each of the ($\Delta = m - n - 1$) intermediate steps introduces an independent integral over the entire action domain $\mathcal{A}$, turning the calculation into an $\Delta$-fold (hyper-)integral whose effective cost grows as $\mathcal{O}(|\mathcal{A}|^{\Delta})$. Thus, the volume of the decision sub-space expands exponentially with the gap $\Delta$.

## A.4  PROOF OF THEOREM 1: CONVERGENCE OF THE TDP SAMPLING

Given a TDP $z$, based on Equation (9), the total uncertainty at step $t$ is:

$$\hat{g}_t(z) = H(\boldsymbol{y}_t|\boldsymbol{y}_{1:t-1},\boldsymbol{x}) + \sum_i^{t-1}\widehat{PMI}(\boldsymbol{y}_t;\boldsymbol{y}_i^{(k)}|\boldsymbol{y}_{i+1:t-1}^{(k)},\boldsymbol{x})\Big), \quad (15)$$

Taking the expectation of Equation (15), we obtain $\mathbb{E}_{z\sim\mathcal{Z}}[\hat{g}_t(z)] = H(\boldsymbol{y}_t|\boldsymbol{x})$. With independent TDPs sampled $\mathcal{Z} = \{z_1, z_2, \cdots\}$, then

$$\hat{H}(\mathcal{P}_{TDP}) = \frac{1}{|\mathcal{Z}|}\sum_z^Z\sum_t^{T_z}\hat{g}_t(z),$$

where $T_z$ is the length of the trajectory $z$. Similarly, taking the expectation over $\mathcal{Z}$, we obtain

$$\mathbb{E}_{z\sim\mathcal{Z}}[\hat{H}(\mathcal{P}_{TDP})] = \sum_t^T H(\boldsymbol{y}_t|\boldsymbol{x}) = H(\mathcal{P}). \quad (16)$$

Thus, we show the estimator is unbiased. With the law of large numbers, we have

$$\hat{H}(\mathcal{P}_{TDP}) = \frac{1}{|\mathcal{Z}|} \sum_z^Z G(z) \rightarrow \mathbb{E}_{z \sim \mathcal{Z}}[G(z)] = H(\mathcal{P}),$$

with $N \rightarrow \infty$, where $G(z) = \sum_t^{T_z} \hat{g}_t(z)$.

### A.5 PROOF OF THEOREM 2: CONVERGENCE OF THE PMI APPROXIMATION

We start from the definition of the PMI conditioned on $\boldsymbol{x}$: $PMI(\boldsymbol{y}_t; \boldsymbol{y}_{t-1}^{(k)}|\boldsymbol{x}) = \log \frac{p(\boldsymbol{y}_t|\boldsymbol{y}_{t-1}^{(k)}, \boldsymbol{x})}{p(\boldsymbol{y}_t|\boldsymbol{x})}$, where $\boldsymbol{y}_{t-1}^{(k)}$ is a realization. According to the local smoothness assumption, for any given $\boldsymbol{y}_{t-1}'$ sufficiently close to the given $\boldsymbol{y}_{t-1}^{(k)}$, it must hold that $p(\boldsymbol{y}_t|\boldsymbol{y}_{t-1}', \boldsymbol{x}) \approx p(\boldsymbol{y}_t|\boldsymbol{y}_{t-1}^{(k)}, \boldsymbol{x})$. Consider the marginalization over $\boldsymbol{y}_{t-1}$:

$$p(\boldsymbol{y}_t|\boldsymbol{x}) = \int p(\boldsymbol{y}_t|\boldsymbol{y}_{t-1}', \boldsymbol{x})p(\boldsymbol{y}_{t-1}'|\boldsymbol{x})d\boldsymbol{y}_{t-1}'. \tag{17}$$

Under the smoothness assumption, within the kernel radius around $\boldsymbol{y}_{t-1}^{(k)}$, we can write:

$$p(\boldsymbol{y}_t|\boldsymbol{x}) \approx \int p(\boldsymbol{y}_t|\boldsymbol{y}_{t-1}^{(k)}, \boldsymbol{x})K_\tau(dist(\boldsymbol{y}_{y_{t-1}}, \boldsymbol{y}_{t-1}^{(k)}))p(\boldsymbol{y}_{t-1}|\boldsymbol{x})d\boldsymbol{y}_{t-1} \quad \succ \textit{Local Smoothness Assumption}$$

$$= p(\boldsymbol{y}_t|\boldsymbol{y}_{t-1}^{(k)}, \boldsymbol{x}) \int K_\tau(dist(\boldsymbol{y}_{y_{t-1}}, \boldsymbol{y}_{t-1}^{(k)}))p(\boldsymbol{y}_{t-1}|\boldsymbol{x})d\boldsymbol{y}_{t-1}$$

$$\approx p(\boldsymbol{y}_t|\boldsymbol{y}_{t-1}^{(k)}, \boldsymbol{x}) \sum_i^N K_\tau(dist(\boldsymbol{y}_{t-1}^{(i)}, \boldsymbol{y}_{t-1}^{(k)})) \quad \succ \textit{MC Approximation}$$

$$= \hat{p}(\boldsymbol{y}_t|\boldsymbol{x}) \tag{18}$$

It is shown that as the sampling number $N \rightarrow \infty$, $\hat{p}(\boldsymbol{y}_t|\boldsymbol{x}) \rightarrow p(\boldsymbol{y}_t|\boldsymbol{x})$, thus $\widehat{PMI}(\boldsymbol{y}_t|\boldsymbol{y}_{t-1}, \boldsymbol{x}) \rightarrow PMI(\boldsymbol{y}_t|\boldsymbol{y}_{t-1}, \boldsymbol{x})$.

### A.6 NEIGHBORHOOD-WEIGHTED AVERAGE IN EQUATION (7)

Computing the exact marginal $p_{\boldsymbol{\theta}}(\boldsymbol{y}_t|\boldsymbol{x})$ requires integrating over all possible trajectories leading to $\boldsymbol{y}_{t-1}$, which is intractable due to the exponential size of the decision space. The neighborhood-weighted average provides an efficient MC-based approximation by leveraging local smoothness in the model's conditional distribution $p_{\boldsymbol{\theta}}(\boldsymbol{y}_t|\boldsymbol{y}_{t-1}, \boldsymbol{x})$ (akin to kernel density estimation), and is widely accepted under mild continuity assumptions.

Equation (7) retains localized inter-step dependency by conditioning on semantically similar samples from the previous step. Specifically, it estimates the marginal $p_{\boldsymbol{\theta}}(\boldsymbol{y}_t|\boldsymbol{x})$ by spreading from the conditional $p_{\boldsymbol{\theta}}(\boldsymbol{y}_t|\boldsymbol{y}_{t-1}, \boldsymbol{x})$, using a neighborhood-weighted average over sampled $\boldsymbol{y}_{t-1}$, reflecting how variations in the prior decision impact the distribution at the current step. Although this does not explicitly integrate over the entire decision history, it preserves localized decision influence critical for uncertainty propagation. For other prior decisions $\boldsymbol{y}_{1:t-2}$, the influence of earlier steps is embedded in the samples of $\boldsymbol{y}_{t-1}$. Each $\boldsymbol{y}_{t-1}^{(n)}$ is generated as part of a full trajectory $\boldsymbol{y}_{1:t-1}^n$, meaning its semantic content implicitly reflects past decisions. Therefore, the approximation does not discard all past information; instead, it utilizes the semantic proximity of these $\boldsymbol{y}_{t-1}$ samples to account for the cumulative effect of preceding decisions. The kernel weighting in Equation (7), controlled by the hyperparameter $\tau$, assigns higher weights to $\boldsymbol{y}_{t-1}^{(n)}$ values that are closer to the anchor $\boldsymbol{y}_{t-1}^{(k)}$. This ensures that the approximation of $p(\boldsymbol{y}_t|\boldsymbol{x})$ is more sensitive to local neighborhoods, effectively capturing how semantically similar or dissimilar prior decisions affect the uncertainty at the current step.

While richer modeling of inter-step dependencies is possible (e.g., via trajectory-level variational inference), such approaches introduce substantial computational overhead. Our goal is to provide a general-purpose, efficient, and scalable UQ estimator, and the proposed neighborhood-weighted strategy strikes a strong balance between fidelity and feasibility. We will leave the advanced sampling and approximation strategies as future work.

Table 5: The results of using the NLI model as the semantic similarity measurement.

| Method | $d$ | AgentBench-OS | StrategyQA | HotpotQA |
|--------|-----|---------------|------------|----------|
| UProp | fuzzy matching | 0.762 | 0.629 | **0.539** |
| UProp | Deberta-large-mnli | **0.767** | **0.635** | 0.537 |

## A.7 MEASURING DISTANCE BETWEEN LLM AGENTIC DECISIONS

In our decision-making environments, at each decision step, LLMs are prompted to provide a ***Reasoning*** output, then followed by an ***Action***. Though the *Reasoning* output is long and versatile, the generated *Action* is usually pre-defined to be short and concise, such as SEARCH(<keyword>) and LOOKUP(<keyword>) in the ReAct agent. Moreover, considering the decision is largely represented by *Action*, the distance between *Actions* becomes an effective measurement of the decision distance. In this way, string fuzzy matching is an efficient method to measure the distance between short actions. Existing work usually applies auxiliary models such as Natural Language Inference (NLI) (Kuhn et al., 2023) model and embedding models (Duan et al., 2024a).

Although we choose fuzzy matching as the distance measurement (due to its efficiency and suitability for short action lengths in multi-step decision-making scenarios), our method can be conveniently extended to more advanced semantic similarity or natural language inference measurements. To demonstrate this, we replace the fuzzy matching with the Deberta-large-mnli (He et al., 2020) model to predict the entailment between two long sentences, which is proven to be effective in comparing the semantics between reasoning responses. We conduct experiments on GPT-4.1-nano and the results are summarized in Table 5. It is worth noting that we reduce the sampling numbers (both step sampling numbers and trajectory sampling numbers) to 5 for computational efficiency. It is shown that replacing with more advanced Deberta-large-mnli benefits Uprop in general. This proves that UProp is a flexible framework that could be easily generalized to handle long responses.

## B EXPERIMENTAL SETUP

### B.1 AGENTBENCH-OPERATING SYSTEM BENCHMARK

The Operating System (OS) split in AgentBench (Liu et al., 2023) enables LLMs to interact with and operate within real operating system (OS) environments through terminal commands, which presents both an exciting opportunity and a significant challenge. It evaluates LLMs in genuine bash environments (specifically, Ubuntu Docker containers using a suite of human-written tasks. These tasks require either deterministic responses—such as querying the number of users without home directories—or the execution of command sequences to achieve practical objectives—such as recursively setting all directory files to read-only, excluding specific users. In this split, there are 144 questions in the testing set, with the expected interaction turns to be 1200.

**Evaluation Metric**: The performance is evaluated by ***Success Rate*** (SR): a task is deemed as a success if the LLM returned answer matches the ground-truth label.

We follow the AgentBench codebase[1] and use the official one-shot demonstration for each OS task. The one-shot demonstration and one test task demonstration are provided as follows:

---

[1]https://github.com/THUDM/AgentBench

918
919

**One-Shot Demonstration**

**USER**
You are an assistant that will act like a person, I'will play the role of linux(ubuntu) operating system. Your goal is to implement the operations required by me or answer to the question proposed by me. For each of your turn, you should first think what you should do, and then take exact one of the three actions: "bash", "finish" or "answer".
1. If you think you should execute some bash code, take bash action, and you should print like this:
Think: put your thought here.
Act: bash
"'bash # put your bash code here "'
2. If you think you have finished the task, take finish action, and you should print like this:
Think: put your thought here.
Act: finish
3. If you think you have got the answer to the question, take answer action, and you should print like this:
Think: put your thought here.
Act: answer(Your answer to the question should be put in this pair of parentheses)
If the output is too long, I will truncate it. The truncated output is not complete. You have to deal with the truncating problem by yourself. Attention, your bash code should not contain any input operation. Once again, you should take only exact one of the three actions in each turn.
Now, my problem is:
tell me how many files are in the directory "/etc"?
**LLM Agent**
Think: To count the files in /etc, I need to print all the files in it.
Act: bash
"'bash ls /etc "'
**USER**
The output of the OS: cpi cron.hourly fuse.conf iproute2 lvm networkd-dispatcher protocols selinux tmpfiles.d [truncated because the output is too long]
**LLM Agent**
Think: The output has been truncated because it is too long, so I need to count files by script directly.
Act: bash
"'bash ls -1 /etc — wc -l "'
**USER**
The output of the OS: 220
**LLM Agent**
agent Think: Now I get the answer, it is 220.
Act: answer(220)
**USER**
Now, I will start a new problem in a new OS. My problem is: *<Test Question>*

955
956

**Test Task Demonstration**

**USER**
Tell me how many files can be executed in PATH.
**LLM Agent**
Think: To count the executable files in PATH, I need to list all directories in PATH and count the executable files in each directory.
Act: bash
"'bash for dir in $(echo $PATH — tr ':' ' '); do echo -n "$dir: " find $dir -maxdepth 1 -type f -executable — wc -l done "'
**USER**
The output of the OS:
echo -n "$dir: " > find $dir -maxdepth 1 -type f -executable — wc -l > done /usr/local/sbin: 0 /usr/local/bin: 0 /usr/sbin: 109 /usr/bin: 446 /sbin: 0 /bin: 0
**LLM Agent**
Think: The total number of executable files in PATH is the sum of the counts in each directory.
Act: bash
"'bash echo $((0 + 0 + 109 + 446 + 0 + 0)) "' user The output of the OS is empty. agent Think: The total number of executable files in PATH is 555.
Act: answer(555)

## B.2 MULTI-STEP REASONING: HOTPOTQA AND STRATEGYQA

**HotpotQA** is a question-answering dataset featuring multi-hop questions: each question needs support from multiple facts. For instance, the question "*What country of origin does House of Cosbys and Bill Cosby have in common?*" requires facts regarding "House of Cosbys" and "Bill Cosby". We follow the ReAct (Yao et al., 2023) experimental setting and randomly select 500 questions from the test split and prompt LLMs in a "Reasoning-Action-Observation" manner:

---

**Prompt Template**

Solve a question answering task with interleaving Thought, Action, Observation steps. Thought can reason about the current situation, and Action can be three types: (1) Search[entity], which searches the exact entity on Wikipedia and returns the first paragraph if it exists. If not, it will return some similar entities to search. (2) Lookup[keyword], which returns the next sentence containing keyword in the current passage. (3) Finish[answer], which returns the answer and finishes the task. Here are some examples. <***Few-Shot Demonstration***>

---

**Few-Shot Demonstrations (HotpotQA)**

**Question**: What is the elevation range for the area that the eastern sector of the Colorado orogeny extends into?
**Thought 1**: I need to search Colorado orogeny, find the area that the eastern sector of the Colorado orogeny extends into, then find the elevation range of the area.
**Action 1**: Search[Colorado progeny]
**Observation 1**: The Colorado orogeny was an episode of mountain building (an orogeny) in Colorado and surrounding areas.
**Thought 2**: It does not mention the eastern sector. So I need to look up eastern sector.
**Action 2**: Lookup[eastern sector]
**Observation 2**: (Result 1 / 1) The eastern sector extends into the High Plains and is called the Central Plains progeny.
**Thought 3**: The eastern sector of Colorado orogeny extends into the High Plains. So I need to search High Plains and find its elevation range.
**Action 3**: Search[High Plains]
**Observation 3**: High Plains refers to one of two distinct land regions:
**Thought 4**: I need to instead search High Plains (United States).
**Action 4**: Search[High Plains (United States)]
**Observation 4**: The High Plains are a subregion of the Great Plains. From east to west, the High Plains rise in elevation from around 1,800 to 7,000 ft (550 to 2,130 m).[3]
**Thought 5**: High Plains rise in elevation from around 1,800 to 7,000 ft, the answer is 1,800 to 7,000 ft.
**Action 5**: Finish[1,800 to 7,000 ft]

---

**StrategyQA** is also a popular multi-hop reasoning benchmark, which requires steps for answering the question. It includes 2,780 examples, with a strategy question, decompositions, and evidence paragraphs within each example. Similar to HotpotQA, we also randomly select 200 questions and prompts LLMs in the ReAct manner with the following few-shot demonstrations.

Table 6: Evaluate the accuracy of `UProp` from hallucination detection.

| Model | PPL | LS | PE | SE | Deg | SD | sentSAR | UProp |
|---|---|---|---|---|---|---|---|---|
| GPT-4.1-Nano | 0.766 | 0.766 | 0.781 | **0.795** | 0.759 | 0.802 | 0.788 | 0.790 |
| Gemma-2-27b-it | 0.762 | 0.711 | 0.800 | 0.785 | 0.725 | 0.770 | 0.777 | **0.837** |
| DeepSeek-V3 | 0.777 | 0.722 | **0.793** | 0.777 | 0.722 | 0.769 | 0.777 | 0.786 |
| **Average** | 0.769 | 0.733 | 0.791 | 0.786 | 0.735 | 0.781 | 0.781 | **0.804** |

Table 7: Evaluate the F1 of `UProp` from hallucination detection.

| Model | PPL | LS | PE | SE | Deg | SD | sentSAR | UProp |
|---|---|---|---|---|---|---|---|---|
| GPT-4.1-Nano | 0.407 | 0.529 | 0.531 | 0.533 | 0.421 | 0.542 | 0.508 | **0.592** |
| Gemma-2-27b-it | 0.333 | 0 | 0.542 | 0.591 | 0.372 | 0.367 | 0.583 | **0.676** |
| DeepSeek-V3 | 0.481 | 0.477 | 0.500 | 0.440 | 0.477 | 0.452 | 0.481 | **0.620** |
| **Average** | 0.407 | 0.335 | 0.524 | 0.521 | 0.423 | 0.454 | 0.524 | **0.629** |

---

**Few-Shot Demonstrations (StrategyQA)**

**Question**: Is Mixed martial arts totally original from Roman Colosseum games?
**Thought 1**: Mixed martial arts (MMA) does have some similarities to the ancient Roman games held in the Colosseum, where gladiators would fight to the death as a form of entertainment. However, there are also distinct differences between the two, such as rules, regulations, and cultural contexts.
**Action 1**: Search[Roman Colosseum games]
**Observation 1**: The Roman Colosseum, also known as the Flavian Amphitheater, was a huge arena used for gladiatorial contests, animal hunts, executions, re-enactments of famous battles, and dramas. The events held at the Colosseum were brutal displays of power and entertainment for the ancient Romans.
**Thought 2**: I need to further check mixed martial arts.
**Action 2**: Lookup[Mixed martial arts]
**Observation 2**: Mixed martial arts is a full-contact combat sport that allows a wide variety of fighting techniques and skills from a mixture of other combat sports to be used in competition. It involves striking and grappling techniques, both standing and on the ground.
**Thought 3**: While both the Roman Colosseum games and modern mixed martial arts involve combat sports, MMA is a regulated sport with rules and safety measures that were not present in the ancient gladiatorial contests. The styles of fighting and the purposes of the two are different, with MMA focusing more on competitive sportsmanship rather than the spectacle of bloodshed and death seen in the Roman games.
**Action 3**: Finish[No]

---

For both HotpotQA and StrategyQA, we use the Wikipedia engine and the environment that takes the retrieval command from LLM Agents and returns the required information.

## C EXTENDED EXPERIMENTAL RESULTS

### C.1 EVALUATION FROM HALLUCINATION DETECTION

we calculate the accuracy and F1 performance of hallucination detection with baselines and UProp. Specifically, we split 20% of the tasks as the validation set to determine the threshold. We apply each method on the validation set and obtain the threshold that maximizes the accuracy of hallucination detection. We then apply this threshold to the rest 80% examples for accuracy and F1 calculation. Results are summarized in Tables 6 and 7. It is shown that UProp achieves superior hallucination detection performance under the evaluation of accuracy and F1.

### C.2 AGGREGATING BASELINES WITH RMS

In Table 8, we compare `UProp` with baselines aggregated by Rooted Mean Square (RMS). RMS aggregation is mainly used to address "outlier" trajectories such as exceptionally large steps and/or

large uncertainties. It is shown that RMS aggregation is worse and simple averaging (Table 1) and `UProp` is significantly better than it.

Table 8: AUROC results over AgentBench-Operating System and StrategyQA benchmarks. For single-turn baseline UQ methods, uncertainties are aggregated by **_RMS_** over all steps.

| Models | Success Rate | PPL | LS | PE | SE | Deg | SD | sentSAR | UProp (ours) |
|---|---|---|---|---|---|---|---|---|---|
| **Benchmark**: AgentBench-Operating System | | | | | | | | | |
| GPT-4.1-Nano | 0.307 | 0.710 | 0.761 | 0.768 | 0.754 | 0.762 | 0.765 | 0.769 | **0.781** |
| GPT-3.5-Turbo | 0.275 | 0.722 | 0.739 | 0.772 | 0.756 | 0.752 | 0.739 | 0.774 | **0.791** |
| Gemma-2-27b-it | 0.289 | 0.731 | 0.639 | 0.750 | 0.739 | 0.653 | 0.755 | 0.754 | **0.814** |
| DeepSeek-V3 | 0.310 | 0.704 | 0.621 | 0.711 | 0.693 | 0.631 | 0.691 | 0.705 | **0.767** |
| Qwen2.5-72B-Instruct | 0.508 | 0.604 | 0.614 | 0.695 | 0.668 | 0.627 | 0.644 | 0.641 | **0.704** |
| **Average** | 0.338 | 0.694 | 0.675 | 0.739 | 0.722 | 0.685 | 0.719 | 0.729 | **0.771** |
| **Benchmark**: StrategyQA | | | | | | | | | |
| GPT-4.1-Nano | 0.691 | 0.516 | 0.505 | **0.551** | 0.506 | 0.520 | 0.502 | 0.539 | 0.544 |
| GPT-3.5-Turbo | 0.611 | 0.607 | 0.435 | **0.620** | 0.608 | 0.438 | 0.601 | 0.530 | 0.604 |
| Gemma-2-27b-it | 0.777 | 0.714 | 0.607 | 0.682 | 0.648 | 0.623 | 0.653 | 0.578 | **0.766** |
| DeepSeek-V3 | 0.790 | 0.578 | 0.552 | 0.557 | 0.557 | 0.572 | 0.574 | 0.460 | **0.607** |
| Qwen2.5-72B-Instruct | 0.796 | 0.500 | 0.509 | 0.573 | 0.579 | 0.514 | 0.560 | 0.496 | **0.617** |
| **Average** | 0.733 | 0.583 | 0.521 | 0.597 | 0.580 | 0.533 | 0.578 | 0.521 | **0.628** |

### C.3 UNCERTAINTY PERCENTAGE

In Figure 6, we provide the detailed uncertainty percentage at each model and benchmark.

### C.4 UPROP IN LONGER SEQUENTIAL DECISION-MAKING

To mitigate the bias inherent in comparing performance across trajectories of varying lengths—where increased length correlates with higher difficulty and inconsistent baseline success rates—we adopt the _Excess AUARC_ metric. Standard area-based metrics are often scale-incomparable when the underlying difficulty of the inference groups differs. Therefore, we quantify the marginal improvement of our uncertainty estimator over a blind baseline. Formally, Excess AUARC is defined as:

$$\text{Excess AUARC} = \text{AUARC}_{\text{method}} - \text{AUARC}_{\text{random}}, \tag{19}$$

where $\text{AUARC}_{\text{method}}$ represents the performance using the proposed uncertainty quantification (UQ) estimator to prioritize rejection, and $\text{AUARC}_{\text{random}}$ denotes the performance of a random rejection policy, which is equivalent to the model's base success rate (accuracy). By subtracting this baseline, we isolate the specific contribution of the UQ ranking quality from the model's intrinsic predictive capability, yielding a scale-consistent and unbiased metric for comparing trajectory groups of heterogeneous lengths.

## D BROADER ADAPTATION: COMPARISON WITH DIVERSE BASELINES

To verify the generality of `UProp` beyond our main setting, we compare against a broad spectrum of uncertainty baselines: (i) logit-based and semantic-consistency methods computed on a single greedy trajectory or the final answer; (ii) trajectory-based variants that aggregate uncertainty over full rollouts; and (iii) last-step decision baselines that only use the final-step uncertainty. Across these families, UProp consistently delivers strong AUROC, often outperforming the strongest baseline within each family.

**Evaluating baselines on a single greedy trajectory and last decision.** We compare against Perplexity (PPL), Mean Token Entropy (MeanTE), Max Token Entropy (MaxTE), and G-NLL, computed on the single greedy trajectory (ST) and on the final answer (FA). As shown in Table 9, `UProp` outperforms these baselines on AgentBench-OS, HotpotQA, and StrategyQA.

**Trajectory-based baselines** We next compare step-level measures with their trajectory-level counterparts, where uncertainty is aggregated over full rollouts. Table 10 shows that trajectory-based

Table 9: Logit-based & semantic-consistency baselines vs. UProp (GPT-4.1-nano).

| Method | AgentBench-OS | HotpotQA | StrategyQA |
|---|---|---|---|
| PPL + ST | 0.734 | 0.605 | 0.538 |
| PPL + FA | 0.738 | 0.619 | 0.527 |
| MeanTE + ST | 0.734 | 0.605 | 0.538 |
| MeanTE + FA | 0.738 | 0.619 | 0.527 |
| MaxTE + ST | 0.651 | 0.596 | 0.500 |
| MaxTE + FA | 0.666 | 0.626 | 0.475 |
| G-NLL + ST | 0.724 | 0.622 | 0.515 |
| G-NLL + FA | 0.763 | 0.644 | 0.528 |
| **UProp** | **0.781** | **0.651** | **0.544** |

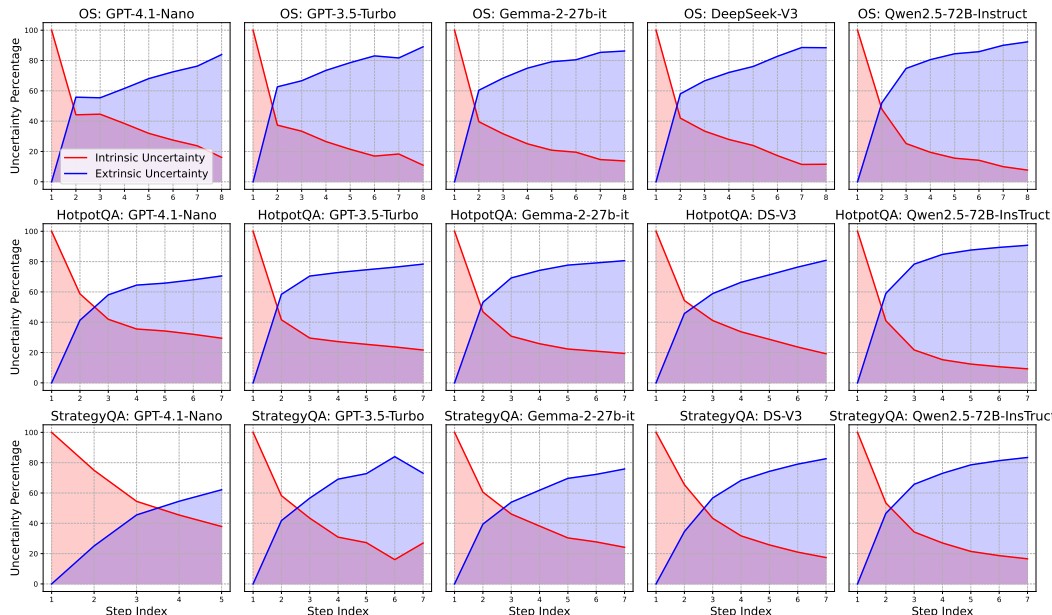

Figure 6: Detailed uncertainty percentage at each decision step.

baselines are generally weaker than step-based ones; importantly, UProp remains competitive or stronger across datasets and models.

Table 10: Step- vs. trajectory-based baselines and UProp.

| (AgentBench-OS) | PE(step) | PE(traj) | SE(step) | SE(traj) | UProp |
|---|---|---|---|---|---|
| GPT-4.1-Nano | 0.768 | 0.736 | 0.770 | 0.763 | **0.781** |
| GPT-3.5-Turbo | 0.782 | 0.730 | 0.765 | 0.745 | **0.791** |
| (StrategyQA) | PE(step) | PE(traj) | SE(step) | SE(traj) | UProp |
| GPT-4.1-Nano | 0.542 | 0.539 | 0.503 | 0.529 | **0.544** |
| GPT-3.5-Turbo | 0.623 | 0.573 | **0.611** | 0.608 | 0.604 |

# E  BROADER DISCUSSION

## E.1  PERMUTATION-INVARIANT TASKS

Permutation-invariant tasks mean the execution order of intermediate decisions within a decision trajectory doesn't affect the final outcome. We first conceptually illustrate that in permutation-

invariant tasks, the variance of a decision-making step consists of two components: 1) model's confidence in the usefulness or relevance of decision; 2) implicit probability choosing this particular permutation.

We denote by $x$ the task or instruction, $\mathcal{Y} = \{y_1, y_2, \cdots, y_T\}$ the set of intermediate decisions needed to solve $x$, $\pi = (y_{\pi(1)}, y_{\pi(2)}, \cdots, y_{\pi(T)})$ a permutation (ordering) of these decisions, and $p_\theta(y_t|y_{<t}, x)$ the model probability of decision $y_t$ at step $t$ conditioned on prior decisions. Then, in a permutation-invariant task, we can decompose the model's probability of choosing $y_t$ at step $t$ as:

$$p_\theta(y_t|y_{<t}, x) \propto p_{\theta,useful}(y_t|x) p_{\theta,perm.}(\pi_t|\mathcal{Y}),$$

where $p_{\theta,useful}(y_t|x)$ reflects the model's confidence in the usefulness or relevance of decision $y_t$ for solving task $x$, independent of position in the sequence, and $p_{\theta,perm.}(\pi_t|\mathcal{Y})$, reflects the implicit probability the model assigns to choosing this particular permutation/order, i.e., how likely it is to select $y_t$ at position $t$ among all valid orderings of $\mathcal{Y}$.

The implicit permutation probability appears in both correct and incorrect permutation-invariant tasks, which cancels the overestimation of uncertainty in the correct outcome (as incorrect outcomes also experience this overestimation due to the existence of $p_{\theta,perm.}(\pi_t|\mathcal{Y})$). Thus, the variance is still an effective metric for the UQ of permutation-invariant tasks.

## F   THE USE OF LARGE LANGUAGE MODELS (LLMS)

For improved clarity and readability, we used OpenAI GPT-4o strictly as an editing aid. Its function was limited to correcting grammar, refining style, and polishing language, much like conventional grammar-checking tools or dictionaries. The model was not involved in generating scientific content or ideas, and its use remains in line with common standards for manuscript preparation.

