# OpenReview forum: "UProp: Investigating the Uncertainty Propagation of LLMs in Multi-Step Decision-Making"
_ICLR.cc/2026/Conference — ICLR 2026 Conference Withdrawn Submission_

### Official Review · Reviewer_oBFz · 2025-10-28

**Soundness:** 3
**Presentation:** 3
**Contribution:** 3
**Rating:** 6
**Confidence:** 3

**Summary:**

This paper addresses the problem of uncertainty quantification in large language models within multi-step decision-making scenarios. It proposes an information-theoretic framework that decomposes uncertainty into ​intrinsic uncertainty​ and ​extrinsic uncertainty. The authors introduce the ​UProp method, which efficiently estimates extrinsic uncertainty by sampling from a ​trajectory-dependent decision process (TDP)​​ and approximating ​pointwise mutual information (PMI)​, thereby tackling the challenge of the inherent non-computability of extrinsic uncertainty. Experiments conducted on benchmarks including ​AgentBench-OS, HotpotQA, and StrategyQA​ validate the superiority of UProp, demonstrating AUROC improvements of 2.3% to 11%. Further analyses on sampling efficiency and applications such as selective prediction are also provided.

**Strengths:**

1. Framework Novelty. It is the first to decompose multi-step decision-making uncertainty into Intrinsic Uncertainty (IU) and Extrinsic Uncertainty (EU), providing a principled framework.
2. Theoretical Soundness and Methodological Innovation. The convergence proofs and the local smoothness assumption enhance the method's credibility. UProp addresses the high computational complexity of Mutual Information (MI) estimation through Trajectory-Dependent Decision Process (TDP) sampling and Pointwise Mutual Information (PMI) approximation, effectively balancing efficiency with accuracy.
3. Comprehensive experiments. The evaluation covers realistic scenarios such as operating system interactions and multi-hop question answering, and includes comparisons with multiple baselines, making the results highly convincing.

**Weaknesses:**

1. High Sampling Dependency.​​ The TDP sampling process requires multiple LLM invocations, which may introduce inference latency issues.
2. Dependence on Theoretical Assumptions.​​ The local smoothness assumption may not always hold in highly uncertain decision chains.
3. Lack of Theoretical Boundary Analysis.​​ The study fails to explicitly specify the upper bound of the impact of PMI estimation errors on the overall uncertainty quantification (UQ) results.

**Questions:**

1. How does the estimation error of Extrinsic Uncertainty (EU) propagate as the number of steps increases?​ Is there an analysis of an upper bound for this error?​
2. In the context of extremely long decision sequences, does extrinsic uncertainty exhibit cumulative drift?​
3. In large-scale tasks, particularly when the decision space expands rapidly, the computational burden of Monte Carlo sampling may become non-negligible. How can the algorithm be optimized to maintain efficiency when handling larger-scale decision-making tasks, especially for application scenarios with high real-time requirements?

---

> ### Author Response · Authors · 2025-11-25
> **Official Rebuttal**
>
> > W1: High Sampling Dependency.​​ The TDP sampling process requires multiple LLM invocations, which may introduce inference latency issues.
>
> We thank you for your comment. We would like to mention that efficiency is an open issue for sampling-based UQ methods. In the LLM multi-step decision-making scenario, UProp avoids exhaustive exploration of the output space by using Trajectory-Dependent Decision Processes (TDP), which convert intractable global estimation into local Pointwise Mutual Information (PMI) estimation. This design ensures that the method scales **linearly** with the number of samples rather than exponentially with the decision tree.
>
> While baselines like Semantic Entropy (SE) and sentSAR are also sampling-based, they incur significant "Semantic Overhead" due to their reliance on heavy auxiliary models (e.g., DeBERTa-large for NLI in SE or RoBERTa for semantic similarity in SAR ). UProp avoids this bottleneck by operating directly on intrinsic probabilities and lightweight string matching. To demonstrate this advantage, we profiled the real latency of Gemma-2-27b-it on the AgentBench-OS task using NVIDIA A40 GPUs. The results are summarized below:
>
> | Method | #n | #z | Semantic Overh. (s) | AUROC |
> | --- | --- | --- | --- | --- |
> | Deg | 10 |10 | <5| 0.76 |
> | SE | 10 |10 | > 20 | 0.75 |
> | sentSAR | 10 | 10 | >40 | 0.75 |
> | UProp (ours) | 2 | 10 | <5 | 0.78 |
> | UProp (ours) | 10 | 2 | <5 | 0.79 |
> | UProp (ours) | 10 | 10 | <5 | 0.81 |
>
> (Semantic overh. means the overhead introduced by semantic consistency/similarity evaluation; we denote by $n$ the number of samples per turn and $z$ the number of trajectories, sampling-based UQ methods require $n \times z$ additional generation calls).
>
> We show that UProp is efficient compared to other state-of-the-art UQ methods and is more **sampling-efficient** (as also highlighted in Figure 2): with much fewer samples, UProp significantly outperforms baselines. We will further investigate optimization techniques to enhance this efficiency in future work.
>
> > W2: Dependence on Theoretical Assumptions.​​ The local smoothness assumption may not always hold in highly uncertain decision chains.
>
> We thank you for your comment. We acknowledge that the Local Smoothness Assumption is central to our PMI approximation. We justify its validity and robustness in highly uncertain chains through three key arguments:
> 1. **Theoretical Necessity**This assumption is not an arbitrary constraint but a fundamental property required for any kernel-based density estimation (effectively assuming Lipschitz continuity). As noted in Section 3.3, this assumption is standard in the theoretical analysis of LLMs (e.g., Malinovskii et al., 2024)1. Without this local regularity, any attempt to generalize uncertainty from finite Monte Carlo samples to the broader decision space would be mathematically intractable.
> 2. **Kernel-Based Implementation** We implement this assumption using a Gaussian Kernel weighted by semantic distance. This design ensures that the "smoothness" is only enforced within a strictly defined neighborhood. If a decision chain is "highly uncertain" to the point of chaos (where $y_{t-1}$ and $y'_{t-1}$ are far apart), the kernel weight approaches zero, naturally preventing the propagation of irrelevant information. The framework does not assume global smoothness, but rather local consistency, which holds true for modern LLMs where semantic meaning is generally robust to minor token variations.
>
> Our empirical results validate that this assumption holds sufficiently well in practice. We will include more discussion in our manuscript.

---

> ### Author Response · Authors · 2025-11-25
> **Official Rebuttal**
>
> > W3&Q1&Q2: Lack of Theoretical Boundary Analysis.​​ How does the estimation error of Extrinsic Uncertainty (EU) propagate as the number of steps increases?​ Is there an analysis of an upper bound for this error?​In the context of extremely long decision sequences, does extrinsic uncertainty exhibit cumulative drift?​
>
> We thank you for the insightful comment. We address the propagation of EU estimation error and the issue of cumulative drift through our theoretical bounds and normalization mechanism.
> 1. **Linear Error Propagation (Upper Bound Analysis)** Based on Equation (9), our estimator aggregates uncertainty via summation, not multiplication. The total uncertainty is calculated as the sum of pointwise terms over the trajectory steps $T$: $\sum_{t=1}^{T} (\text{IU}_t + \text{EU}_t)$. Consequently, the error propagation is additive. If the estimation error for a single PMI term is bounded by $\epsilon$ (guaranteed to be small by the convergence in Theorem 2 2), the upper bound of the total error for a sequence of length $T$ is $\mathcal{O}(T \cdot \epsilon)$. This linear bound is crucial as it ensures that estimation errors do not explode exponentially (as they might in multiplicative frameworks), making the method theoretically stable even for longer sequences.
> 2. **Addressing Cumulative Drift via Normalization** Regarding "cumulative drift" in extremely long sequences, we explicitly recognize this phenomenon in the paper as "Step Length Bias", where longer trajectories naturally accumulate higher total uncertainty (entropy). To prevent this natural accumulation, we employ Step Length-Normalization using the factor $\lambda_z$. As defined in Equation (9), we normalize the cumulative uncertainty by $\lambda_z = \sum_{t}^{T_z} \sigma_t$. This effectively scales the total uncertainty relative to the trajectory length. This normalization ensures that the uncertainty metric remains comparable across varying lengths, preventing the "cumulative drift" from invalidating the metric in extremely long decision sequences.
> 3. We further quantify the performance of UProp over short and long trajectories individually, to explore whether UProp experiences a performance degradation at extremely long trajectories. For a fair comparison, we calculate Excess AUARC to quantify the pure gain over trajectories with various numbers of steps:
> $$\text{Excess AUARC} = \text{AUARC}_{\text{Method}} - \text{AUARC}_{\text{Random}}$$. We calculate Excess AUARC on Gemma-2-27b-it and DeepSeek-V3 over the strategyqa dataset. Results are summarized as follows:
>
> | Model | 2 steps| 6 steps |10 steps|
> | --- | --- | --- | --- |
> | Gemma-2-27b-it | 0.098 |0.043 | 0.062 |
> | DeepSeek-V3 | 0.037 | 0.031 | 0.029 |
>
> It is shown that Excess AUARC is stable at different numbers of steps, indicating that UProp is empirically effective on questions requiring longer trajectories.
>
> > Q3: In large-scale tasks, particularly when the decision space expands rapidly, the computational burden of Monte Carlo sampling may become non-negligible. How can the algorithm be optimized to maintain efficiency when handling larger-scale decision-making tasks, especially for application scenarios with high real-time requirements?
>
> We thank you for your comment. We acknowledge that Monte Carlo (MC) sampling introduces overhead. However, UProp is inherently designed to mitigate the "exploding decision space" problem, and we propose several optimization strategies for real-time scenarios:
>
> 1. **Linear Complexity via TDP** UProp addresses the "decision spaces expand rapidly" issue by avoiding exhaustive tree search. Instead, we utilize TDP to sample linear-spanning decision paths rather than exploring the exponential tree. This ensures that our computational complexity scales linearly with the number of trajectories ($Z$) and steps ($T$), rather than exponentially ($N^T$).
> 2. **Tunable Efficiency for Real-Time Constraints** UProp offers a direct "knob" to trade off precision for speed. Our empirical analysis of *Sampling Efficiency* (Figure 2) demonstrates that UProp outperforms baselines even when restricted to just 2 trajectory samples ($Z=2$). In high real-time requirement scenarios, practitioners can set $Z$ and $N$ to minimum values (e.g., 2-3), drastically reducing latency while retaining significant uncertainty detection capabilities.

---

> > ### Comment · Reviewer_oBFz · 2025-11-28
> >
> > Thanks to the authors for their systematic rebuttal. The rebuttal is clear overall, and it has addressed most of the questions. But, in my opinion, there are still three key issues in the rebuttal that haven’t been fully resolved:
> > 1. The case for the robustness of the local smoothness assumption is weak. The authors only talked about the theoretical mechanism of the "Gaussian kernel" but didn’t provide weight distribution data or error curves for high-uncertainty decision chain scenarios. They also didn’t add comparative experiments on task subsets with semantic mutations, so we can’t confirm if it works in extreme cases.
> > 2. the verification of long-sequence drift is incomplete. The new Excess AUARC experiment only compares performance across different steps, it doesn’t measure how much the EU estimate drifts from the true value in the same long sequence.
> > 3. The error upper bound isn’t practical. The authors didn’t specify the range of single-step errors in typical tasks, nor explain how sample count relates to error control. So there’s not enough useful guidance for setting parameters.
> > Based on the current rebuttal, I will still keep my earlier rating: "6: marginally above the acceptance threshold."

---

> > > ### Author Response · Authors · 2025-12-03
> > >
> > > > 1. The case for the robustness of the local smoothness assumption is weak. The authors only talked about the theoretical mechanism of the "Gaussian kernel" but didn’t provide weight distribution data or error curves for high-uncertainty decision chain scenarios. They also didn’t add comparative experiments on task subsets with semantic mutations, so we can’t confirm if it works in extreme cases.
> > >
> > > We thank you for your comment. As suggested, we design a test focusing on the most volatile cases. Specifically, we filtered the StrategyQA benchmark to isolate the Top 20% tasks with the highest IU. These are the decision chains where the model is most likely to be uncertain, representing the "extreme cases" requested by the reviewer. In terms of semantic mutation, to simulate the local neighborhood explored by our Gaussian kernel, we applied perturbations to the input prompts by randomly swapping two words, and then sampled model decision with greedy generation.
> > >
> > > We measured the stability of the LLM's output distribution before and after mutation. Despite the high intrinsic uncertainty of these tasks, the semantic mutations did not cause catastrophic shifts in the output distribution. We observed a significantly small KL divergence between the original and mutated predictive distributions (0.1 in average), which is at the same level as the KL divergence over the 20% lowest IU subset (0.06). The high degree of overlap confirms that the decision space satisfies the local smoothness property even in high-uncertainty regions. Small perturbations in the semantic space (the inputs) lead to small, predictable changes in the probability space (the outputs), validating the use of the Gaussian kernel for spreading probability mass in UProp's PMI estimation.
> > >
> > >
> > >
> > > > 2. How much the EU estimate drifts from the true value in the same long sequence.
> > >
> > > We agree with the reviewer that measuring drift against the "true" Extrinsic Uncertainty is ideal. However, as noted in the paper (Section 3.1), calculating the true MI requires integrating over the entire exponential decision space. This is computationally **intractable** for LLMs.
> > >
> > > Instead, we provide a theoretical error analysis for the "Drift". We define "Drift" as the accumulated estimation error over a sequence of length $T$. Let $\epsilon$ be the upper bound of the approximation bias for a single PMI term (dependent on sample count $N$).The unnormalized Extrinsic Uncertainty at step $T$ is the sum of previous interactions4:$$EU_T = \sum_{i=1}^{T-1} PMI(y_T; y_i)$$Consequently, the total drift at step $T$ is bounded by the sum of per-step errors:
> > > $$Drift(T) = | \widehat{EU}_{T} - {EU}\_{true} | \leq \sum\_{i=1}^{T-1} \epsilon = (T-1)\epsilon$$
> > >  If without intervention, error accumulates linearly with sequence length ($O(T)$).
> > >
> > > However, we **explicitly incorporate a Step Length-Normalization factor $1/\lambda_z$ (Equation 9)**. Since $\lambda_z$ scales proportionally with the sequence length $T_z$, the final effective drift becomes:$$\text{Effective Drift} \propto \frac{\text{Drift}(T)}{T} \approx \frac{(T-1)\epsilon}{T} \to \epsilon$$
> > >
> > > Therefore, while long sequences introduce cumulative errors ($O(T)$), our normalization strategy ensures the average error per step remains bounded. This ensures that UProp remains robust even in long-horizon tasks, as supported by our experiments on AgentBench (which involves long decision trajectories).

---

> > > > ### Author Response · Authors · 2025-12-03
> > > >
> > > > > 3. The authors didn’t specify the range of single-step errors in typical tasks, nor explain how sample count relates to error control.
> > > >
> > > > We thank the reviewer for your insightful comment. As suggested, we derive the explicit relationship between sample counts $(Z, N)$ and the estimation error to provide concrete guidance for parameter selection.
> > > > Let's denote by $H$ the true uncertainty, $\hat{H}\_{Z,N}$ the uncertainty estimated by UProp, and $\mathbb{E}[\hat{H}\_{Z,N}]$ the expected value of the estimator over infinite trajectory samples. To quantify the error, we decompose the total estimation error into two distinct components using the triangle inequality:$$| \hat{H}_{Z,N} - H | \leq \underbrace{| \hat{H}\_{Z,N} - \mathbb{E}[\hat{H}\_{Z,N}] |}\_{\text{(I) Statistical Error}} + \underbrace{| \mathbb{E}[\hat{H}\_{Z,N}] - H |}\_{\text{(II) Approximation Bias}}.$$
> > > > Here, **Statistical Error** (Dependent on $Z$) represents the variance caused by sampling a finite number of Trajectory-Dependent Decision Processes (TDPs). **Approximation Bias** (Dependent on $N$) represents the systematic error (bias) introduced by estimating the Pointwise Mutual Information (PMI) using a finite number of samples $N$. Even with infinite trajectories, this bias exists because we approximate the true probability density using a kernel over $N$ local neighbors2.
> > > >
> > > > Since UProp calculates the mean over $Z$ independent TDPs, the Central Limit Theorem applies. For a decision process variance $\sigma^2\_{TDP}$, the statistical error is bounded with high probability ($1-\delta$) by:$$\text{Error}\_{stat} \leq z_{1-\delta/2} \frac{\sigma\_{TDP}}{\sqrt{Z}}$$(Proof Sketch: This follows standard Monte Carlo convergence rates, where the standard error of the mean decreases in proportion to $1/\sqrt{Z}$).
> > > > In terms of bounding Approximation Bias ($N$), it stems from the kernel density estimation in our PMI approximation. Under the local smoothness assumption (Theorem 2), the bias of the kernel density estimator with bandwidth $h$ and sample size $N$ scales as $O(N^{-\alpha})$.$$\text{Bias}\_{approx} \leq \frac{C}{N^{\alpha}}$$ (Proof Sketch: As $N$ increases, the kernel density estimate converges to the true marginal distribution $p(y_t|x)$, reducing the systematic distance between the estimated PMI and the true PMI). Combining these bounds, the total error upper bound is:$$|\hat{H}_{Z,N} - H| \leq \frac{A}{\sqrt{Z}} + \frac{B}{N^{\alpha}}$$This indicates that error control requires a simultaneous increase in trajectory samples ($Z$) to reduce variance and per-step samples ($N$) to reduce bias, explaining how sample count relates to error control.

---

### Official Review · Reviewer_MkZL · 2025-10-29

**Soundness:** 3
**Presentation:** 3
**Contribution:** 3
**Rating:** 6
**Confidence:** 3

**Summary:**

The paper proposes UProp, a method to quantify uncertainty in Large Language Models for multi-turn question-answering and sequential decision-making—critical for judging when to trust LLM outputs. UProp decomposes uncertainty into two parts: Intrinsic Uncertainty(IU) and Extrinsic Uncertainty(EU). And Uprop calculates the EU through Trajectory-Dependent Pointwise MI since directly calculating EU is intractable. Experiments across multi-turn benchmarks (e.g., AgentBench-OS, HotpotQA) and state-of-the-art LLMs show UProp achieves more accurate uncertainty quantification than single-turn UQ baselines.

**Strengths:**

1. The paper is clearly written and aims to tackle the more realistic and challenging setting of multi-turn QA, providing a rigorous, information-theoretic foundation for it.
2. UProp proposes a novel perspective for quantifying uncertainty in multi-turn question-answering. Its "PMI in TDP" method offers an efficient solution by converting the computationally expensive direct estimation of Mutual Information (MI) into a tractable process.

**Weaknesses:**

1. Although UProp employs sampling to approximate distributions, this strategy introduces non-trivial computational costs, resulting in slower inference speeds and increased resource consumption. These factors may limit the method's applicability in real-time or large-scale deployment scenarios.
2. The method heavily depends on a smoothness assumption. While intuitive, this assumption lacks theoretical grounding and may not hold in all practical conditions, potentially compromising the framework's generalizability.
3. UProp is primarily evaluated on commonsense reasoning tasks with definitive answers. Consequently, its effectiveness and contributions remain unclear for other problem categories, such as those involving open-ended generation or creative tasks.
4. There are some typos in the paper, e.g, Figure 1, Section 5.

**Questions:**

1. The results on StrategyQA (Table 1) show a higher success rate for Qwen2.5 but a much lower AUROC compared to Gemma-2. Please discuss this result. Does it indicate that the estimated uncertainty does not always directly correlate with the final task success rate?
2. Could the authors elaborate on the potential of Uprop in more exploratory environments (e.g., creative writing, simulations), where leveraging uncertainty is crucial for guiding the agent's actions to achieve higher performance?
3. Is it possible to integrate the EU component with other single-turn UQ methods, and if so, what would be the effect?
4. UProp employs ReAct as the reasoning method. How well would it perform with other reasoning methods?
5. Quantifying output uncertainty in LLMs is a critical consideration, and recent work, PlanU [1], explores a similar direction. PlanU extends uncertainty quantification to the context of LLMs in reasoning tasks, aligning with the objectives of UProp. However, PlanU also introduces the additional aspect of considering environmental uncertainty. It would be valuable to further explore the distinctions between UProp and PlanU, especially in terms of how each approach handles their implications for uncertainty quantification.

[1] PlanU: Large Language Model Reasoning through Planning under Uncertainty. NeurIPS 2025

**Details Of Ethics Concerns:**

No.

---

> ### Author Response · Authors · 2025-11-25
> **Official Rebuttal**
>
> > W1: Although UProp employs sampling to approximate distributions, this strategy introduces non-trivial computational costs.
>
> We thank you for your comment. We would like to mention that efficiency is an open issue for sampling-based UQ methods. In the LLM multi-step decision-making scenario, UProp avoids exhaustive exploration of the output space by using Trajectory-Dependent Decision Processes (TDP), which convert intractable global estimation into local Pointwise Mutual Information (PMI) estimation. This design ensures that the method scales **linearly** with the number of samples rather than exponentially with the decision tree.
>
> While baselines like Semantic Entropy (SE) and sentSAR are also sampling-based, they incur significant "Semantic Overhead" due to their reliance on heavy auxiliary models (e.g., DeBERTa-large for NLI in SE or RoBERTa for semantic similarity in SAR ). UProp avoids this bottleneck by operating directly on intrinsic probabilities and lightweight string matching. To demonstrate this advantage, we profiled the real latency of Gemma-2-27b-it on the AgentBench-OS task using NVIDIA A40 GPUs. The results are summarized below:
>
> | Method | #n | #z | Semantic Overh. (s) | AUROC |
> | --- | --- | --- | --- | --- |
> | Deg | 10 |10 | <5| 0.76 |
> | SE | 10 |10 | > 20 | 0.75 |
> | sentSAR | 10 | 10 | >40 | 0.75 |
> | UProp (ours) | 2 | 10 | <5 | 0.78 |
> | UProp (ours) | 10 | 2 | <5 | 0.79 |
> | UProp (ours) | 10 | 10 | <5 | 0.81 |
>
> (Semantic overh. means the overhead introduced by semantic consistency/similarity evaluation; we denote by $n$ the number of samples per turn and $z$ the number of trajectories, sampling-based UQ methods require $n \times z$ additional generation calls.).
>
> We show that UProp is efficient compared to other state-of-the-art UQ methods and is more **sampling-efficient** (as also highlighted in Figure 2): with much fewer samples, UProp significantly outperforms baselines. We will further investigate optimization techniques to enhance this efficiency in future work.
>
>
> > W2: The method heavily depends on a smoothness assumption. While intuitive, this assumption lacks theoretical grounding and may not hold in all practical conditions.
>
> We thank you for your comment. We would like to mention that the assumption of local smoothness is not an ad-hoc invention but a standard property in the analysis of deep neural networks. As noted in our paper (Section 3.3), this assumption is widely used in existing LLM theoretical analysis. Formally, this is equivalent to assuming Lipschitz continuity of the predictive distribution $p_\theta(y_t|y_{t-1}, x)$ with respect to the context $y_{t-1}$. This is a fundamental requirement for any kernel-based density estimation or interpolation method to guarantee convergence. Without this standard assumption, generalizing from finite Monte Carlo samples to a continuous distribution would be mathematically impossible for any sampling-based method.
>
> From the empirical validation perspective, our extensive empirical results suggest that UProp consistently outperforms 7 baselines across diverse benchmarks (AgentBench-OS and StrategyQA) and multiple model families (GPT-4, DeepSeek, Llama-2). This provides strong experimental evidence that the smoothness assumption holds sufficiently well in practical LLM decision landscapes.
>
>
> > W3&Q2: UProp is primarily evaluated on commonsense reasoning tasks with definitive answers. Could the authors elaborate on the potential of Uprop in more exploratory environments?
>
> We thank you for your insightful comment. We include Mind2Web[1], a new exploratory environment, as the benchmark of UProp. We conduct experiments on Mind2Web with the same protocols (e.g., temperature, number of samplings) as other benchmarks over GPT-3.5-Turbo and GPT-4.1-Nano. Results are summarized as follows:
>
> | Model | PE | SE | sentSAR | Deg | SD | UProp |
> | --- | --- | --- | --- | --- | --- | --- |
> | GPT-3.5-Turbo | 0.57 | 0.59 | 0.59 | 0.56 | 0.57 | **0.61** |
> | GPT-4.1-Nano | 0.52 | 0.56 | 0.54 | 0.51 | 0.49 | **0.59** |
> | avg. | 0.54 | 0.58 | 0.56 | 0.53 | 0.53 | **0.60** |
>
> We show that UProp is still effective in the challenging exploratory environments. We will include more discussion in our manuscript.
>
> Reference:
> [1] Deng, Xiang, et al. "Mind2web: Towards a generalist agent for the web." Advances in Neural Information Processing Systems 36 (2023): 28091-28114.
>
> > W3: There are some typos in the paper, e.g, Figure 1, Section 5.
>
> We thank you for your proofreading. We will fix them in our next revision.

---

> ### Author Response · Authors · 2025-11-25
> **Official Rebuttal**
>
> > Q1: The results on StrategyQA (Table 1) show a higher success rate for Qwen2.5 but a much lower AUROC compared to Gemma-2. Please discuss this result. Does it indicate that the estimated uncertainty does not always directly correlate with the final task success rate?
>
> We thank you for your insightful comment. We would like to claim that UProp achieving low AUROC on QWen2.5 compared to Gemma-2 does not indicate that UProp uncertainty fails to correlate with success; rather, it indicates that "accuracy" and "calibration" (the ability to rank correctness) are distinct properties.
>
> We attribute this phenomenon to the **"Confidently Wrong"** nature often observed in capable, instruction-tuned models. Fundamentally, AUROC quantifies the capability to rank correctness, a task that relies heavily on a distinct differentiation between the uncertainty scores of correct and incorrect generations. When a model is "confidently wrong," it maintains low intrinsic uncertainty even when making mistakes (hallucinating). This behavior drastically reduces the margin between correct and incorrect uncertainty scores, making the ranking task significantly harder and resulting in a lower AUROC. We statistically verified the response probabilities and found that QWen2.5’s correct and incorrect examples share very similar distributions, compared with Gemma-2.
>
>
> > Q3: Is it possible to integrate the EU component with other single-turn UQ methods, and if so, what would be the effect?
>
> We thank you for your insightful comment. UProp is a general uncertainty propagation framework for LLM multi-step decision-making. It is compatible with other UQ methods by replacing the IU calculation from Predictive Entropy (PE) with single-turn UQ methods. For instance, we provide the comparison between UProp (w/ PE) and UProp (w/ G-NLL):
>
> |Model	|Method	|AgentBench-OS	|HotpotQA	|StrategyQA|
> | --- | --- | --- | --- | --- |
> |GPT-4.1-Nano	|UProp (w/ PE, default)	|**0.781**	|**0.651**	|**0.544**|
> |GPT-4.1-Nano	|UProp (w/ G-NLL)	|0.742	|0.620	|0.532|
> |GPT-3.5-turbo	|UProp (w/ PE, default)	|0.791	|**0.721**	|**0.604**|
> |GPT-3.5-turbo	|UProp (w/ G-NLL)	|**0.793**	|0.704	|**0.604**|
> |Deepseek-V3	|UProp (w/ PE, default)	|**0.767**	|0.684	|0.607|
> |Deepseek-V3	|UProp (w/ G-NLL)	|0.744	|**0.703**	|**0.615**|
>
> It is shown that replacing the IU calculation from PE to G-NLL improves UProp in some cases. This demonstrates that UProp is a general framework that could be built upon existing single-sequence UQ methods.
>
> > Q4: UProp employs ReAct as the reasoning method. How well would it perform with other reasoning methods?
>
> UProp is a generalizable uncertainty-propagation framework that can be integrated with any reasoning paradigm. We use ReAct primarily because (1) it is one of the most widely adopted approaches for multi-step decision-making, and (2) its simple “Reason → Action → Observation” structure offers a clean setting to isolate modeling effects and analyze model behaviors. In future work, we plan to extend UProp to more advanced reasoning methods, such as Reflexion.
>
> > Q5: Explore the distinctions between UProp and PlanU, especially in terms of how each approach handles their implications for uncertainty quantification.
>
> We thank the reviewer for highlighting the relevance of PlanU. We acknowledge that both works address the critical challenge of reasoning under uncertainty in LLMs. However, there are fundamental distinctions in the objective and theoretical framework.
>
> 1. **Objective**: The primary goal of UProp is UQ as a metric for correctness. We aim to produce a scalar score (IU + EU) that correlates with correctness to enable Selective Prediction (rejecting hallucinations). Differently, the primary goal of PlanU is Planning and Control. It is an MCTS-based decision-making method designed to maximize task rewards. It quantifies uncertainty (via quantile distributions ) specifically to guide the exploration-exploitation trade-off (using the UCC score ) within a search tree, rather than to serve as a post-hoc reliability metric for the final answer.
>
> 2. **Theoretical Framework**: UProp is grounded in Information theory. We decompose uncertainty into Intrinsic (entropy of the current step) and Extrinsic (Mutual Information from the history). We measure the information flow between decision steps. Differently, PlanU models the return (cumulative reward) of a state-action pair as a Quantile Distribution rather than a mean value. It uses Quantile Regression to estimate the variance of future rewards, not the information-theoretic confidence of the generated tokens for environmental uncertainty.
>
> We will include PlanU as a related work and provide more discussions.

---

### Official Review · Reviewer_4e7F · 2025-10-29

**Soundness:** 2
**Presentation:** 3
**Contribution:** 3
**Rating:** 6
**Confidence:** 2

**Summary:**

This paper addresses the challenge of Uncertainty Quantification (UQ) for LLMs in multi-step decision-making. The authors propose UProp, a method that decomposes the total uncertainty into Intrinsic Uncertainty (IU) from the current decision and Extrinsic Uncertainty (EU) inherited from previous steps, and efficiently estimates the computationally challenging EU.

**Strengths:**

1. The research topic is valuable, targeting the under-explored uncertainty propagation in LLM multi-step decision-making (a critical issue for reliable agentic LLMs), the work aligns with current academic and industrial needs.
2. UProp avoids complex auxiliary models or exponential space exploration, relying on lightweight sampling and approximation—striking a good balance between computational efficiency and estimation accuracy.

**Weaknesses:**

- The reliance on a simple random sampling strategy for trajectory candidates lacks justification. The lack of discussion or ablation studies on sampling strategies leaves open the question of how sensitive UProp's performance is to the quality and diversity of the sampled trajectories.
- The paper quantifies extrinsic uncertainty but does not verify its correlation with final decision error rates (e.g., whether higher extrinsic uncertainty correlates with more frequent multi-step reasoning errors)
- Unaddressed performance in long-sequence scenarios, no experiments or discussions are provided on UProp’s behavior in multi-step decisions with over 10 steps (e.g., whether step-length normalization fully mitigates length bias, or if PdMI approximation error accumulates), limiting the method’s applicability to complex long-horizon tasks.

**Questions:**

- How was the sampling strategy designed? Could the estimator's accuracy be significantly improved by employing a more intelligent sampling technique that prioritizes informative or diverse decision paths?
- Is there a measurable correlation between the estimated extrinsic uncertainty and the actual error rate in multi-step decisions? Could the authors provide experimental evidence (e.g., error rate vs. EU scatter plots) to strengthen the practical utility of UProp?
- How does UProp perform in long-sequence decision tasks? If estimation accuracy degrades with more steps, what strategies could be applied to maintain stability and mitigate bias?

---

> ### Author Response · Authors · 2025-11-25
> **Official Rebuttal**
>
> > W1&Q1: The reliance on a simple random sampling strategy for trajectory candidates lacks justification. How was the sampling strategy designed? Could the estimator's accuracy be significantly improved by employing a more intelligent sampling
>
> We thank you for your insightful comment. Our "simple random sampling" (specifically, multinomial sampling based on the model's probability $p_\theta$) is chosen not for simplicity, but for statistical correctness in Monte Carlo (MC) estimation.
>
> 1. **Justification for Model-Based Sampling** Our goal is to estimate the information-theoretic quantities (Entropy and Mutual Information) of the current model's distribution
> $p_\theta(y|x)$
> . The standard definition of predictive entropy is
> $H(y|x) = \mathbb{E}(y \sim p_{\theta})[-logp_\theta(y|x)]$
>  To approximate this expectation unbiasedly via MC integration, samples must be drawn according to the underlying distribution $p_\theta$. Employing a strategy that artificially prioritizes "diverse" or "informative" paths (deviating from $p_\theta$) would introduce bias. While such methods (e.g., diverse beam search) are excellent for generation (finding the best answer), they would distort the uncertainty estimation unless rigorous Importance Sampling weights are applied, which might be unstable in high-dimensional LLM spaces.
> 2. **Empirical Sensitivity Analysis** Regarding the sensitivity to sampling, in Section 4.3, we evaluated UProp's performance by varying the number of trajectories $Z$ and per-step samples $N$ (Figure 2). Results demonstrate that UProp is highly robust, outperforming baselines even with minimal sampling (e.g., $Z=2$). This indicates that the estimator is not hypersensitive to the specific set of trajectories, provided they are drawn from the model's distribution.
> Moreover, we provide the performance of UProp over various temperatures. Results are summarized as follows (Gemma-2-27b-it on AgentBench-OS):
>
> | Model | Temperature | AUROC |
> | --- | --- | --- |
> | UProp | 0.5 | 0.79 |
> | UProp | 0.8 (default) | 0.81 |
> | UProp | 1.2| 0.82 |
>
> A higher temperature encourages more diverse sampling trajectories, whereas a lower temperature produces more consistent ones. Our results show that UProp is robust to this variation in sampling diversity. We will include this analysis in the manuscript and investigate more intelligent sampling strategies for UProp in future work.
>
> > W2&Q2: The paper quantifies extrinsic uncertainty but does not verify its correlation with final decision error rates.
>
> Thank you for the insightful comment. Following your suggestion, we compute extrinsic uncertainty as the final uncertainty for each question and evaluate its Pearson correlation with correctness (0/1). The results are summarized below:
>
> | Model | Pearson |
> | --- | --- |
> | Gemma-2-27b-it | -0.44 |
> | DeepSeek-V3 | -0.37 |
> | Qwen2.5-72B-Instruct-Turbo | -0.36 |
>
> It is shown that the extrinsic uncertainty is significantly negatively correlated to the correctness of answers, which indicates that extrinsic uncertainty is an effective indicator and uncertainty estimator. In Section 4.4 (Figure 4), as suggested, we plot the EU vs. correctness scatter, as well as the linear convergence. It is shown that extrinsic uncertainty is negatively correlated with correctness.
>
> > W3&Q3: Unaddressed performance in long-sequence scenarios. How does UProp perform in long-sequence decision tasks? If estimation accuracy degrades with more steps, what strategies could be applied to maintain stability and mitigate bias?
>
> Thank you for the insightful comment. Directly comparing AUROC across trajectories with different numbers of steps can indeed introduce bias, because longer trajectories are inherently more challenging than shorter ones. This variation in difficulty places AUROC values on incomparable scales across trajectory groups. To enable a fair comparison, we instead report Excess AUARC for trajectories of varying lengths. Excess AUARC measures the marginal improvement a UQ estimator provides over blind guessing:
> $$
> Excess AUARC = AUARC_{Method} - AUARC_{Random}
> $$
>
> $AUARC_{Method}$
> is the performance using an estimator to intelligently reject incorrect answers first.
> ${AUARC}_{Random}$
> is simply the Success Rate (Accuracy) of the model1. It represents the performance if you rejected answers randomly (i.e., your UQ estimator had zero intelligence). This subtraction isolates the quality of the ranking provided by the UQ method, yielding a scale-consistent and unbiased metric across trajectory groups of different lengths. We compute Excess AUARC for Gemma-2-27B-IT and DeepSeek-V3 on the StrategyQA dataset. Results are summarized below:
>
> | Model | 2 steps| 6 steps |10 steps|
> | --- | --- | --- | --- |
> | Gemma-2-27b-it | 0.098 |0.043 | 0.062 |
> | DeepSeek-V3 | 0.037 | 0.031 | 0.029 |
>
> It is shown that Excess AUARC is stable at different numbers of steps, indicating that UProp is still effective on questions requiring longer trajectories.

---

### Official Review · Reviewer_i8hd · 2025-10-30

**Soundness:** 3
**Presentation:** 2
**Contribution:** 2
**Rating:** 4
**Confidence:** 3

**Summary:**

The paper introduces an estimate method to estimate the LLM sequential decision uncertainty. First, the paper decomposes LLM sequential decision uncertainty into two parts, including internal uncertainty intrinsic to the current decision and extrinsic uncertainty as a Mutual-Information (MI) quantity of how much uncertainty is inherited from preceding decisions. Second, the proposed method as UProp achieves effective extrinsic uncertainty estimation, which converts the direct estimation of MI to the estimation of Pointwise Mutual Information over multiple Trajectory-Dependent Decision Processes. The results show that UProp significantly outperforms existing single-turn Uncertainty Quantification baselines, including sampling efficiency, potential applications, and intermediate uncertainty propagation.

**Strengths:**

1. The theoretical analysis is relatively rigorous.
2. The experimental results show that the algorithm is effective.
3. The method simply estimates the proposed Extrinsic uncertainty in the multi-step decision-making of LLM with the pointwise mutual information by the Monte Carlo sampling.

**Weaknesses:**

1. The innovation of the paper needs further clarification.
2. There is a lack of computational cost analysis of the method.
3. The method may be hard to generalize due to the large output space and hallucination of LLM.

**Questions:**

1. Did you first propose to decompose the uncertainty of decision-making into intrinsic and extrinsic uncertainty with the information theory? If not, please introduce some previous related work. If yes, please provide more evidence.
2. The backbone of LLM model may be additionally selected, including Claude 3 or Gemini 2.5 Pro, which has a powerful ability of decision-making or reasoning.
3. Please add some benchmarks such as complex mathematical reasoning tasks or long-chain knowledge reasoning, etc.
4. After quantifying intrinsic and Extrinsic uncertainties, how can you optimize LLM decisions through ReAct-style prompts?
5. There are some problems with the manuscript. Why are some words underlined? And the symbols such as “➋” are uncommon in the academic papers. The researchers usually use “2)” or “b.” instead.

---

> ### Author Response · Authors · 2025-11-24
> **Official Rebuttal**
>
> > W1&Q1: The innovation of the paper needs further clarification. Did you first propose to decompose the uncertainty of decision-making into intrinsic and extrinsic uncertainty with the information theory? If not, please introduce some previous related work. If yes, please provide more evidence.
>
> We thank you for your insightful comment. We confirm that we propose the first information-theoretic framework specifically tailored for LLM sequential decision-making, which decomposes uncertainty into Intrinsic Uncertainty (IU) and Extrinsic Uncertainty (EU) to model uncertainty propagation. Entropy decomposition is a fundamental mathematical tool and conceptually, our terms parallel the established concepts of aleatoric and epistemic uncertainty[1,2], but with a crucial distinction in the context of sequential agents. Aleatoric uncertainty typically attributes system uncertainty to irreducible data noise, whereas epistemic uncertainty attributes it to the lack of knowledge in model parameters. We develop this decomposition specifically for LLM sequential decision-making to study uncertainty propagation. In this framework, IU captures the internal variance of the current decision dependent on the current state, while EU quantifies the uncertainty introduced by, or inherited from, the variability of preceding decisions.
> Therefore, we extend uncertainty decomposition method to LLM multi-step decision-making scenario. Existing UQ methods primarily focus on single-turn QA and fail to capture this propagation of uncertainty across a trajectory.
>
> Moreover, simply proposing the theory is not enough, as calculating this Mutual Information is computationally intractable due to the exponential decision space. Our contribution includes UProp, an efficient and effective estimator that converts the direct estimation of MI into the estimation of Pointwise Mutual Information (PMI) over Trajectory-Dependent Decision Processes (TDPs), making this theoretical framework practically solvable for large-scale LLMs.
>
> Reference:
>
> [1] Kendall, Alex, and Yarin Gal. "What uncertainties do we need in bayesian deep learning for computer vision?." Advances in neural information processing systems 30 (2017).
>
> [2] Depeweg, Stefan, et al. "Decomposition of uncertainty in Bayesian deep learning for efficient and risk-sensitive learning." International conference on machine learning. PMLR, 2018.
>
> > W2: There is a lack of computational cost analysis of the method.
>
> We thank you for your comment. Similar to other UQ methods, our proposed UProp is a sampling-based UQ method. To rigorously compare costs, we analyze the complexity in terms of both sampling queries and latency:
> 1. Sampling Complexity: Assuming $n$ samples per turn and $z$ trajectories, UProp requires $n \times z$ additional generation calls. This is identical to baseline methods like PE, SE, and SAR, which also require $z \ge 1$ trajectories and multiple samples to estimate uncertainty.
> 2. Computational Overhead: A critical distinction is that baselines such as SE and SAR rely on computationally heavy auxiliary models (e.g., DeBERTa-large or RoBERTa) to calculate semantic entailment or sentence similarity. This introduces an additional complexity of $\mathcal{O}(z \cdot n^2)$. In contrast, UProp is computationally lightweight as it operates directly on token probabilities without external NLI models.
>
> Consequently, despite similar sampling counts, UProp achieves significantly lower real-world latency. We include a summary below comparing sampling efficiency and profiling real latency on NVIDIA A40 GPU to demonstrate this advantage (Gemma-2-27b-it on AgentBench-OS task):
>
> | Method | #n | #z | Semantic Overh. (s) | AUROC |
> | --- | --- | --- | --- | --- |
> | Deg | 10 |10 | <5| 0.76 |
> | SE | 10 |10 | > 20 | 0.75 |
> | sentSAR | 10 | 10 | >40 | 0.75 |
> | UProp (ours) | 2 | 10 | <5 | 0.78 |
> | UProp (ours) | 10 | 2 | <5 | 0.79 |
> | UProp (ours) | 10 | 10 | <5 | **0.81** |
>
> (Semantic overh. means the overhead introduced by semantic consistency evaluation.)
>
> We show that UProp is efficient compared to other UQ methods and is more sampling-efficient (as also highlighted in Figure 2): with much fewer samples, UProp significantly outperform baselines.

---

> ### Author Response · Authors · 2025-11-24
> **Official Rebuttal**
>
> > W3: The method may be hard to generalize due to the large output space and hallucination of LLM.
>
> We thank you for your comment. Resolving "large output space" and "hallucination" are exactly the core contributions of our paper. We respectfully point out that our method was designed specifically because these challenges exist:
>
> 1. **We do not search the "Large Output Space" (The TDP Solution)** If we were estimating global Mutual Information, the reviewer would be correct: it would not generalize. However, our key innovation is UProp, which circumvents the large output space entirely. Uprop converts the intractable global estimation into Pointwise Mutual Information (PMI) estimation over Trajectory-Dependent Decision Processes (TDP). This reduces the complexity from exponential (exploring the whole tree) to linear (sampling specific paths). Our efficiency analysis (Figure 2) proves this generalization capability: UProp outperforms baselines even when sampling as few as 2 trajectories ($Z=2$). This proves we do not need to cover the "large output space" to be effective.
> 2. **Hallucination is the "Signal," not the "Blocker"** The reviewer mentions hallucination as a hindrance. In the context of UQ, hallucination is actually the phenomenon we aim to capture. When an LLM hallucinates, it typically introduces divergence between decision trajectories. Our Extrinsic Uncertainty (EU) term effectively captures this "inherited variability". Our Selective Prediction experiments (Table 3) demonstrate that UProp successfully identifies and filters these hallucinations (rejecting incorrect answers), outperforming 7 state-of-the-art baselines5555. This confirms that the method does not just "survive" hallucination, but it effectively detects it.
>
> > Q2: The backbone of LLM model may be additionally selected, including Claude 3 or Gemini 2.5 Pro, which has a powerful ability of decision-making or reasoning.
>
> We thank you for your comment. As suggested, we use Gemini 2.5 flash and Gemini 2.5 Pro as the new models and conduct experiments on StrategyQA and HotpotQA. Results are summarized as following:
>
> | Model | Dataset | PE | SE | sentSAR | Deg | SD | UProp |
> | --- | --- | --- | --- | --- | --- | --- | --- |
> |Gemini 2.5 Flash | StrategyQA | 0.60 | 0.57 | 0.59 | 0.59 | 0.60 | **0.70** |
> |Gemini 2.5 Pro | StrategyQA | 0.64 | 0.59 | 0.65 | 0.63 | 0.66 | **0.72** |
>
> We show that, our method still significantly outperform baselines on new Gemini models, indicating that UProp could be applied in powerful large reasoning models.
>
> > Q3: Please add some benchmarks such as complex mathematical reasoning tasks or long-chain knowledge reasoning, etc.
>
> We thank you for your insightful comment. As suggested, we conduct experiments on the MuSiQue[1] dataset, which is one of the most popular long-chain question-answering benchamrk over multiple models. Results are summarized as the following:
>
> | Model | PE | SE | sentSAR | Deg | SD | UProp |
> | --- | --- | --- | --- | --- | --- | --- |
> |GPT-3.5-turbo | 0.78 | 0.76 | 0.77 | 0.79 | 0.80 | **0.82** |
> |GPT-4o-nano| 0.75 | 0.75 | 0.72 | 0.60 | 0.72 | **0.78** |
>
> It is shown that our methods substantially outperform baslines over multiple models. We will include more math-related benchmarks such as GSM-Agent in our next revision.
>
> Reference:
> [1] Trivedi, Harsh, et al. "♫ MuSiQue: Multihop Questions via Single-hop Question Composition." Transactions of the Association for Computational Linguistics 10 (2022): 539-554.
>
>
> > Q4: After quantifying intrinsic and Extrinsic uncertainties, how can you optimize LLM decisions through ReAct-style prompts?
>
> We thank you for your comment. To enhance LLM reasoning performance vis uncertainty guidance, we do not update ReAct prompts. Instead, once we obtained the uncertainty estimated by UProp, we follow the selective prediction manner[1,2] and 1) sample the responses multiple times then 2) select the response with the lowest uncertainty as the final answer. This technique is common and effective in improving the performance of LLM decision. In Table 3 and Figure 3, we show it substantially enhances the performance of ReAct agents.
>
> Reference:
> [1] Wang, Xuezhi, et al. "Self-consistency improves chain of thought reasoning in language models." arXiv preprint arXiv:2203.11171 (2022).
> [2] Lin, Zhen, Shubhendu Trivedi, and Jimeng Sun. "Generating with confidence: Uncertainty quantification for black-box large language models." arXiv preprint arXiv:2305.19187 (2023).
>
> > Q5: There are some problems with the manuscript. Why are some words underlined? And the symbols such as “➋” are uncommon in the academic papers. The researchers usually use “2)” or “b.” instead.
>
> Thank you for your careful proofreading. We use underline to highlight our claims. We will remove them and replace “➋” with "2)" in our manuscript.

---

### Public Comment · ~Zhizhao_Liu1 · 2025-11-12

Thank you for this interesting paper. I enjoyed reading it and have a question regarding the formulation in lines 213-221.
You state that: "Conditioned on the realizations within TDP, the mutual information (MI) $I(y_t; y_{t−1}|x)$ over TDP at step t becomes a pointwise mutual information (PMI)." Could you please elaborate on why this is the case?
My confusion stems from the relationship between PMI and KL divergence. According to the standard definition, the PMI is calculated as:
$$PMI(y_t;y_{t-1}=y_{t-1}^{(k)}|x) = \log \frac{p_{\theta}(y_t,y_{t-1}^{k}|x)}{p_{\theta}(y_t|x)p_{\theta}(y_{t-1}^{k}|x)}$$
From my understanding, $p_{\theta}(y_t|y_{t-1}^{(k)},x) \cdot PMI(y_t;y_{t-1}=y_{t-1}^{(k)}|x)$ is equivalent to the KL divergence: $D_{KL}(p_{\theta}(y_t|y_{t-1}^{(k)},x)\parallel p_{\theta}(y_t|x))$

This seems to contradict the equation in your paper, where you state that:

$$PMI(y_t;y_{t-1}=y_{t-1}^{(k)}|x)=D_{KL}(p_{\theta}(y_t|y_{t-1}^{(k)},x)\parallel p_{\theta}(y_t|x))$$

Furthermore, regarding the approximation of MI with PMI, the standard expression for MI is an expectation over the PMI:

$$I(y_t;y_{t-1}|x) = \mathbb{E}{p_{\theta}(y_t,y_{t-1}|x)} \left[ \log \frac{p_{\theta}(y_t,y_{t-1}|x)}{p_{\theta}(y_t|x)p_{\theta}(y_{t-1}|x)} \right]$$

This definition, which involves the joint probability $p_{\theta}(y_t,y_{t-1}^{(k)}|x)$, does not seem to be equivalent to directly using the PMI as an approximation for MI.
Could you please provide a detailed explanation for these points? I want to ensure I haven't misunderstood something. Is it possible that I have made an error in my derivation, or perhaps I am missing some context about the problem? Alternatively, is this a common approximation or technique in uncertainty estimation that I am not aware of?
Thank you for your time and clarification.

---

### Comment · Area_Chair_AxP6 · 2025-11-27

Dear Reviewers,

Thanks for taking the time to review this work. The authors have responded to your reviews. Can you please have a look at the rebuttal and discuss with the authors?

Best Regards,

AC

---

### Note · Authors · 2026-01-28

I have read and agree with the venue's withdrawal policy on behalf of myself and my co-authors.

---

### Meta-Review · Area_Chair_czyA · 2026-01-06

**Summary:**

This paper explores how uncertainty "travels" through the multiple steps an LLM takes when solving a task. It splits uncertainty into two parts: the uncertainty of the current step and the uncertainty inherited from previous steps. While the reviewers found the topic important and the experiments on benchmarks like AgentBench and StrategyQA useful, there are significant concerns regarding the paper’s core contributions. The idea of decomposing uncertainty is not entirely new, and the proposed UProp method relies on heavy sampling, which makes it very slow and expensive for real-world use. Furthermore, the mathematical assumption that the model’s behavior is "smooth" across different word choices is difficult to justify in the discrete world of language.

**Reviewer Concerns:**

During the rebuttal, the authors successfully added experiments on more models and showed that their uncertainty scores correlate well with actual model errors. However, several critical issues remain outstanding. Reviewers were not fully convinced by the "local smoothness" assumption, noting that a single different word can completely change a trajectory, breaking the mathematical logic used here. Additionally, the analysis of how errors grow in very long sequences was seen as "weak" by the most engaged reviewer. The computational cost—requiring many different trajectories to be sampled—remains a major barrier that the rebuttal did not fully solve.

**Reviewer Scores:**

Reviewer i8hd (Score 4) likely would not have changed their score, as they remained skeptical about the overall innovation of the work. While Reviewers 4e7F, MkZL, and oBFz gave the paper a "6" (marginally above), their written comments and the subsequent discussion suggest their confidence was low. Specifically, Reviewer oBFz noted after the rebuttal that the authors' explanations for core assumptions were still "weak."

---

### Decision · Program_Chairs · 2026-01-26

Reject